# Efficient Variable Selection Using Reinforcement Learning for Big Data

## Abstract

Efficient variable selection is crucial for optimizing the performance and interpretability of machine learning models. However, as datasets expand in sample size and dimensionality, traditional methods may encounter computational challenges and accuracy issues. To tackle this problem in the realm of big data, we propose a novel approach referred to as *REinforcement learning for Variable Selection* (REVS) within the Markov Decision Process (MDP) framework. By prioritizing the long-term variable selection accuracy, we propose a dynamic policy to adjust the candidate important variable set, guiding it toward convergence to the true variable set. To enhance computational efficiency, we present an online policy iteration algorithm integrated with temporal difference learning for sequential policy improvement. Our experiments demonstrate superior performance of the method, especially in big data scenarios where it substantially reduces computation time compared to alternative methods. Furthermore, in high-dimensional feature sets with strong correlations, our approach enhances variable selection accuracy by leveraging cumulative reward information from batch data.

## 1 Introduction

The field of Reinforcement Learning (RL) is often concerned with the problem of how to make the optimal sequential decisions in dynamic environments for agents, with the aim of maximizing the cumulative rewards in long terms. In recent years, RL has made significant advances and has shown potential for various applications in scientific domains, such as robotics (Silver et al., 2016; Kalashnikov et al., 2018; Li et al., 2023; Nikkhoo et al., 2023), autonomous driving (Sallab et al., 2017; Chen et al., 2021b) and personalized medicine (Weltz et al., 2022; Gao et al., 2022a). Recently, RL has demonstrated its ability to enhance the efficiency of fundamental problems requiring extensive computational resources. For example, Fawzi et al. (2022) introduced a deep RL approach based on AlphaZero1 that can tackle matrix multiplication of any size efficiently. Wu et al. (2020) proposed model-based RL technique for hyperparameter tuning in complex machine learning algorithms, particularly within the context of large scale datasets. Drawing inspiration from the success of RL in optimizing large scale computational algorithms, our aim is to adapt these techniques to address challenges in supervised learning within the realm of machine learning. In particular, our focus centers on *variable selection* in *big data* by leveraging the power of RL.

### 1.1 Motivations and Related Work

Variable selection plays a pivotal role in optimizing the performance and interpretability of machine learning models. It can be achieved through regularization methods that induce sparsity by controlling the number of non-zero coefficients in the model. In supervised learning, the optimization problem can be formulated as the combination of *loss + penalty*, moderated by a regularization parameter to balance these two components.

Best subset selection, which penalizes the $\ell_0$-based norm of coefficients, is a notable method in this context (Greenshtein, 2006; Raskutti et al., 2011; Zhang et al., 2014; Bertsimas & Van Parys, 2020). However, it is an NP-hard problem and computationally challenging. To approximate solutions to the best subset selection, various methods have been proposed. These include continuous proxies to the $\ell_0$ norm, such as $\ell_1$ norm

(LASSO) (Tibshirani, 1996), mixture penalization of $\ell_1$ and $\ell_2$ norm (Elastic Net) (Zou & Hastie, 2005), and non-convex penalizations such as SCAD (Fan & Li, 2001) and MCP (Zhang, 2010). Nonetheless, in scenarios with *high-dimensional* and *highly-correlated* data, these methods may not effectively recover the sparsity pattern, leading to biased estimations (Zhang & Huang, 2008) and suboptimal performance in terms of accuracy and false discovery rates (Bertsimas et al., 2020). Furthermore, these regularization-based approaches are sensitive to the tuning of hyperparameters. Systematic hyperparameter optimization, especially when coupled with cross-validation in *big data* contexts with millions of observations, presents substantial computational challenges (Yao & Allen, 2020). This computational complexity can significantly impedes the practical deployment of these methods in real-world scenarios.

To address computational challenges in big data, one popular approach is *data reduction* through strategic subsampling. The basic idea is to select the most informative data points to create a smaller dataset that retains most information from the full dataset (Drineas et al., 2006; 2011; Wang et al., 2018). The effectiveness of these approaches critically depends on the choice of sampling probabilities. Unlike uniform sampling, empirical statistical leverage scores from the input covariate matrix are often employed to define non-uniform subsampling probabilities, a technique known as *algorithmic leveraging* (Ma et al., 2015). Furthermore, an information-based subsampling has been proposed to further improve the computation efficiency (Wang et al., 2019). However, subsampling inevitably leads to certain information loss and the data-dependent sampling process can introduce bias. Additionally, above methods primarily address the ordinary least squares problem by using all the features, and are not suitable for variable selection.

In parallel, inspired by the divide-and-conquer strategy, *distributed learning* frameworks have been developed to handle large-scale statistical optimization challenges (Jordan, 2012; Zhang et al., 2013; Chen & Zhou, 2020). These frameworks break down complex tasks into smaller segments that are processed concurrently across multiple computing units, with their results aggregated for the final outcome (Gao et al., 2022b). While this strategy can reduce computation time, its inherent design of executing tasks independently, rather than in a sequence, limits the ability to leverage insights gained from previous stages to enhance subsequent ones. Recently, inspired by the multi-armed bandit and reinforcement learning problems, Yao & Allen (2020); Fan et al. (2020; 2021); Liu et al. (2021) have proposed methods to adaptively select both observations and features, sequentially adjusting the non-sparse variable set using batch data. These approaches have shown good performance in datasets with moderate sample sizes. Our paper focuses on extending these methodologies to scenarios involving large sample sizes, aiming to addressing the challenges from the scalability and efficiency in more extensive data environments.

## 1.2 Research Question

*In the context of the datasets with **large sample sizes** characterized by high-correlated features, how can we efficiently and accurately identify the true non-sparse feature set? Is it feasible to employ a sequential approach for gradually adjusting the non-sparse feature set using batch data, and how can this process be optimized?*

To address above question regarding efficient and accurate variable selection in large datasets, we turn to the principles of RL. RL is a powerful machine learning technique that enables an agent to make sequential decisions in order to maximize the long-term reward. RL problems are often structured under the Markov Decision Process (MDP) framework (Puterman, 2014; Sutton & Barto, 2018). In MDP, the data are collected in a sequential way and can be summarized as the triplet information of state, action and reward: $\{(S_t, A_t, R_t)\}_{0 \le t \le T}$, where $T$ is denoted as the number of stages. One central objective is to learn the optimal policy, $\pi^*$, a strategy enabling decision-makers to choose actions based on the current state at each stage, with the goal of maximizing the expected discounted cumulative reward.

## 1.3 Major Contributions

We propose a novel approach referred to as *REinforcement learning for efficient Variable Selection* (REVS). REVS integrates RL techniques to tackle the variable selection challenge in *big data* contexts. First, we conceptualize the variable selection problem as a *time-homogeneous MDP*, effectively treating it as a policy-driven process. This involves viewing each small data batch, sampled from the large dataset, as a stage in

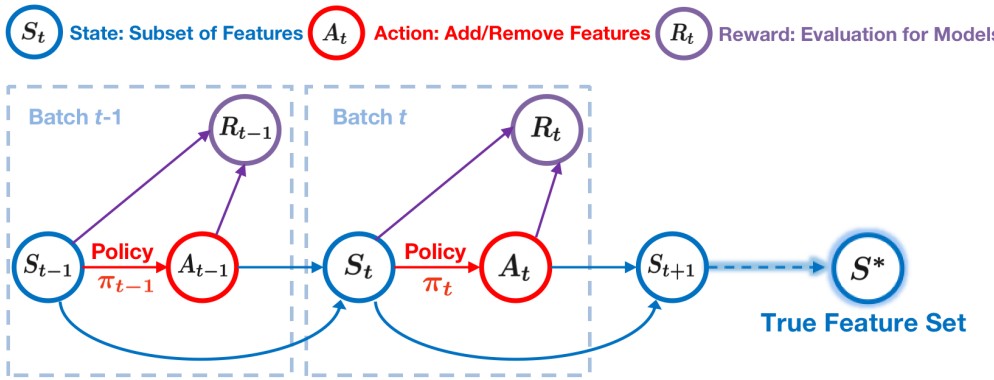

Figure 1: Formulate *variable selection* problem as MDP.

**MDP.** Within each batch $t$, the candidate non-sparse variable set represents our state, $S_t$. Our goal is to dynamically adjust this set by adding or removing variables, or keeping it constant, guided by our evolving estimated policy. Such adjustments lead to the transition to the next state, $S_{t+1}$. To evaluate the efficacy of these changes, we fit the variable sets into a supervised learning model for each stage, with the model's performance serving as the reward. This performance feedback is then used to update our policy through online policy iteration algorithm to iteratively update our policy until it converges. See Figure 1 as an illustration. Our proposal leverages the iterative and adaptive nature of RL and applies these principles to the variable selection process in large datasets.

Second, REVS leverages the crucial RL advantage of balancing exploration and exploitation. Compared with traditional methods such as backward or forward selection (Derksen & Keselman, 1992; Mao, 2004; Zhang, 2008), which may get stuck at local optima due to their greedy nature, our approach employs strategies such as $\epsilon$-greedy for policy updates. While we exploit the existing knowledge about variable sets that have shown promise (*exploitation*), we also intermittently explore new variable combinations that haven't been previously considered (*exploration*). By incorporating the exploration-exploitation trade-off, REVS is designed not just for local variable selection accuracy (*short-term*), but to enhance the overall accuracy of variable selection over batch (*long-term*) in large datasets.

Third, REVS enhances computational efficiency in variable selection for large datasets. By fitting simpler models on smaller data batches under the MDP framework, we reduce the computational burden commonly associated with cross-validation in traditional regularization methods using the entire dataset. Furthermore, within each batch, we propose the quantile navigation approach so that REVS concentrates on a smaller subset of active variables, achieving additional computational efficiency compared to other mini-batch methods that process all variables. Moreover, by utilizing cumulative reward information from each batch, this iterative process of adjusting selected variables with varied training and validation batches inherently mitigates the risk of overfitting and expedites the convergence. In our experiments, REVS demonstrates promising results, showing enhanced efficiency and accuracy compared to traditional variable selection techniques using penalized regression, particularly in scenarios involving big data and high-dimensional datasets with correlated features.

Fourth, compared with prior approaches for feature selection using reinforcement learning (Fard et al., 2013; Rasoul et al., 2021), REVS introduces a more general and efficient framework to address the limitations in big data: (1) We propose a refined MDP formulation that allows both the addition and removal of features at each stage, offering greater flexibility and adaptability while adhering to the Markov assumption, while the action space in previous works is limited to only adding features. (2) Our reward function, based on the improvement in prediction loss, generalizes the framework's applicability to a wide range of supervised learning tasks, surpassing the simpler feature scoring metrics used in earlier methods. (3) To tackle computational challenges, our approach decomposes the optimization problem into manageable steps, achieving significant efficiency gains, particularly for large datasets. (4) The epsilon-greedy algorithm balances exploration and exploitation, reducing overfitting in high-dimensional and correlated data settings. (5) We also provide

theoretical guarantees, including convergence to optimal feature sets and a detailed analysis of the Bellman equation, establishing a strong foundation for the method's robustness and reliability.

## 2 Formulate Variable Selection with MDP

Suppose we have covariates $X = (X_1, X_2, \ldots, X_p)^\intercal$ of dimension $p$, and a response variable $Y$. Assuming observations are independently and identically drawn from the population $(X^\intercal, Y)$, we consider the Generalized Linear Model (GLM, McCullagh & Nelder (1989)):

$$g(\mathbb{E}[Y|X]) = \beta_0 + \boldsymbol{\beta}^\intercal X, \tag{1}$$

where $g(\cdot)$ is a known link function that relates the expected value of $Y$ to the linear predictor $\boldsymbol{\beta}^\intercal X$, $\beta_0$ is the intercept, and $\boldsymbol{\beta}^\intercal = (\beta_1, \ldots, \beta_p)$. The GLM framework can cover a broad spectrum of supervised learning tasks, each characterized by a specific link function. For example, when the identity link function is used, it leads to the linear regression model $Y = \beta_0 + \boldsymbol{\beta}^\intercal X + \epsilon$, where $\epsilon$ is the error term. For binary classification tasks such as logistic regression, GLM utilizes the logit link function, resulting in the model $\log\left(\frac{\mathbb{E}[Y|X]}{1-\mathbb{E}[Y|X]}\right) = \beta_0 + \boldsymbol{\beta}^\intercal X$. In a sparse GLM, it is typically assumed that most regression coefficients $\beta_j$ are 0. The main goal of variable selection is to identify significant variables with non-zero coefficients, and accurately estimate them for predicting $Y$. In settings with complex feature structures and big data, we aim to adapt RL to enhance computational efficiency and variable selection accuracy.

We propose to formulate the problem of variable selection with MDP. A '*stage*' in our context specifically refers to a decision point in the iterative batch process to adjust variables. Each stage involves sampling a small size of batch data from the big dataset. The primary objective is to use the information from each batch of data to sequentially refine the selected variable set. For variable selection, we define the critical elements of *state*, *action*, and *reward* as follows:

### 2.1 State: *Current Selected Variable Set*

Let the state space $\mathcal{S}$ be the power set of features in $X$, i.e., $|\mathcal{S}| = 2^p$. The optimal state, denoted as $\boldsymbol{s}^*$, corresponds to the true variable set with non-zero regression coefficients. The state at each stage $t$, represented by $S_t$, reflects the current selection of variables. At each stage $t$, we randomly sample a batch data with small sample size from the large-scaled data. We use $\bar{S}_t$ to denote all historical triplet information up to that point, along with the current state $S_t$. The goal is to sequentially assess and potentially adjust the selected variable set through actions such as adding significant variables or removing redundant ones, thus guiding the transition from one stage to the next, e.g., $\boldsymbol{s}_1 \to \boldsymbol{s}_2 \to \boldsymbol{s}_3 \to \ldots$, and ultimately converging to the optimal state $\boldsymbol{s}^*$ across a series of batch data.

### 2.2 Action: *Add/Remove One Variable*

To ensure stability in the variable adjustment process, each stage is limited to either adding/removing one variable or maintaining the existing set for the next stage. Specifically, the action space is defined as

$$\mathcal{A} = \left\{ \underbrace{+1, +2, \ldots, +p}_{\text{Add}}, \underbrace{-1, -2, \ldots, -p}_{\text{Remove}}, \underbrace{0}_{\text{Invariant}} \right\}.$$

Here, for $j = 1, 2, \ldots, p$, the action "$+j$" means adding variable $X_j$ to the current state, while the action "$-j$" refers to removing variable $X_j$ from the current state. The action "$0$" signifies maintaining the current variable set for the next state. From the computational perspective, given a state $\boldsymbol{s}$, the total number of feasible actions is $p + 1$, as we cannot add already existing variables in $\boldsymbol{s}$ or remove absent ones.

The transition probability $\mathbb{P}_t : \bar{\mathcal{S}}_t \times \mathcal{A} \to \Delta^{\mathcal{S}}$ governs the shift from the history information $\bar{\boldsymbol{s}}_t \in \bar{\mathcal{S}}_t$ to the next state $\boldsymbol{s}_{t+1} \in \mathcal{S}$ when taking action $a_t \in \mathcal{A}$ at time $t$. Our MDP formulation ensures that $\mathbb{P}_t$ is deterministic, time-homogeneous, and stationary. Specifically, the transition depends solely on the current variable set $\boldsymbol{s}_t$

and action $a_t$, without any time dependency. In particular, we use a single transition function $\mathbb{P}$ to denote it:

$$\mathbb{P}_t[\boldsymbol{s}_{t+1}|\bar{\boldsymbol{s}}_t, a_t] = \begin{cases} \mathbb{I}[\boldsymbol{s}_{t+1} = \{\boldsymbol{s}_t \bigcup X_j\}] & \text{if } a_t = +j, \\ \mathbb{I}[\boldsymbol{s}_{t+1} = \{\boldsymbol{s}_t \backslash X_j\}] & \text{if } a_t = -j, \\ \mathbb{I}[\boldsymbol{s}_{t+1} = \boldsymbol{s}_t] & \text{if } a_t = 0, \end{cases} \tag{2}$$
$$:= \mathbb{P}[\boldsymbol{s}_{t+1}|\boldsymbol{s}_t, a_t].$$

This deterministic and stationary nature aligns with the Markov assumption, affirming the consistency and stationarity of state transitions in our MDP formulation.

### 2.3 Reward: *Evaluation of Model Performance*

We define the immediate reward $R_t$ at each stage $t$ to assess the performance of the current action. Intuitively, a beneficial action is one that moves the current state progressively closer to the optimal state $\boldsymbol{s}^*$, which should provide the best prediction for the response variable $Y$. Our reward function $R : \mathcal{S}_t \times \mathcal{A} \times \mathcal{S}_{t+1} \to \mathbb{R}$ consists of two components: a measure of change in model performance $r_{\text{perf}}$, and a penalty term for the current action $r_{\text{pen}}$ to regulate the number of selected variables:

$$\begin{aligned} r_t &= R(\boldsymbol{s}_t, a_t, \boldsymbol{s}_{t+1}) \\ &= \underbrace{f(\boldsymbol{s}_t, \boldsymbol{s}_{t+1})}_{r_{\text{perf},t}: \text{ Change in model performance}} - \underbrace{\lambda \, \text{sign}(a_t)}_{r_{\text{pen},t}: \text{ Penalty for action}}. \end{aligned} \tag{3}$$

In particular, for linear regression, the first term $r_{\text{perf}}$ can be specified as the decline in *mean square error*, while for logistic regression, it can be the increase in *log-likelihood*, both measuring the improvement in model performance when using variables transitioning from $\boldsymbol{s}_t$ to $\boldsymbol{s}_{t+1}$ due to action $a_t$ to fit the model. The second term is a penalty that decreases the reward by $\lambda$ when new variables are added (*overcoming potential overfitting*) and increases it when variables are removed (*stimulating valid variable selection process*). When $a_t = 0$, the reward $r_t = 0$ since the state remains unchanged. This penalty term effectively acts as a sequential $\ell_0$-based regularization, penalizing the number of selected variables based on the action taken at each stage.

### 2.4 Policy: *A Strategy Guiding the Adjustment of Variable Sets*

In RL, a policy is a sequence of decision rules, $\{\pi_t\}_{t \geq 0}$, guiding the decision-maker on which action to choose at each time $t$. In the context of variable selection, our policy determines how to intelligently adjust the variables in the current state $\boldsymbol{s}_t$ at each stage to progressively move from $\boldsymbol{s}_t$ towards the optimal state $\boldsymbol{s}^*$ as stage progresses. Given that our defined MDP upholds the assumptions of stationary transition and reward, we can only focus on a stationary and time-homogeneous policy class $\Pi$ (Sutton & Barto, 2018). This class is independent of time $t$ and historical information preceding it. A policy $\pi$ within this class satisfies the condition $\pi(a|S_t) = \pi_t(a|\bar{S}_t)$ for any $t$, ensuring consistency in decision-making regardless of the stage. Under a specific policy $\pi \in \Pi$, at each decision point $t$, the action $A_t = a$ is chosen from the action space with a probability dictated by $\pi(a|\boldsymbol{s})$, given the current variable set $S_t$ as $\boldsymbol{s}$. Our policy $\pi$ takes the state value in $\mathcal{S}$ as input and outputs a probability distribution over the action space $\mathcal{A}$:

$$\begin{aligned} \pi(\boldsymbol{s}) = \big(&\pi(+1|\boldsymbol{s}), \pi(+2|\boldsymbol{s}), \dots, \pi(+p|\boldsymbol{s}), \pi(-1|\boldsymbol{s}), \\ &\pi(-2|\boldsymbol{s}), \dots, \pi(-p|\boldsymbol{s}), \pi(0|\boldsymbol{s})\big)^{\mathsf{T}}. \end{aligned} \tag{4}$$

Here, $\pi(+j|\boldsymbol{s})$ and $\pi(-j|\boldsymbol{s})$ represent the probabilities of adding or removing the variable $X_j$, respectively. Specifically, if $X_j \in \boldsymbol{s}$, then $\pi(+j|\boldsymbol{s}) = 0$. Conversely, if $X_j \notin \boldsymbol{s}$, then $\pi(-j|\boldsymbol{s}) = 0$. Meanwhile, $\pi(0|\boldsymbol{s})$ corresponds to the probability of maintaining the current state $\boldsymbol{s}$. This policy framework enables the adaptive and strategic navigation through the variable selection process, balancing the need to explore new variable combinations with the aim of converging towards an optimal variable set for accurate prediction.

The main objective of RL is to identify the optimal policy that yields the highest discounted cumulative reward. Similarly, for any given policy $\pi \in \Pi$ and any initial state $\boldsymbol{s} \in \mathcal{S}$, our value function is defined as $V^\pi(\boldsymbol{s}) = \mathbb{E}^\pi \big[ \sum_{t \geq 0} \gamma^t R(S_t, A_t, S_{t+1}) | S_0 = \boldsymbol{s} \big]$, where $\mathbb{E}^\pi$ denotes the expectation of the trajectory when the

actions are selected according to $\pi$. Here, $\gamma \in (0, 1)$ is denoted as the fixed discounted factor that balances the trade-off between the immediate and long-term rewards. The $Q$-function, denoted as $Q^\pi(\boldsymbol{s}, a)$, is defined as the discounted cumulative reward where the initial state-action pair is $(\boldsymbol{s}, a)$ and then all subsequent actions follow the policy $\pi$: $Q^\pi(\boldsymbol{s}, a) = \mathbb{E}^\pi \big[ \sum_{t=0}^{\infty} \gamma^t R(S_t, A_t, S_{t+1}) | S_0 = \boldsymbol{s}, a_0 = a \big]$.

With a substantial number of i.i.d. observations of covariate and response $(\boldsymbol{x}_i, y_i)$ in big data, our objective is to estimate the optimal policy $\pi^* \in \arg \max V^\pi(\boldsymbol{s})$, which maximizes the expected discounted cumulative reward for each state $\boldsymbol{s} \in \mathcal{S}$. The optimal policy $\pi^*$ characterizes a specific way of sequentially adjusting the variable set. To handle the large data size, we break the dataset into smaller and manageable batches. This enables us to iteratively refine the policy in an online manner, using each batch of data to sequentially update the policy based on insights gained at each stage. The reward function $R$, combining the change in model performance and penalty for controlling redundant variables, guides selection towards the optimal set of variables $\boldsymbol{s}^*$.

Let $\pi_{(t)}$ be the estimated optimal policy under each stage $t$. As the learning progresses, the estimated policy $\pi_{(t)}$ evolves, assigning higher probabilities to actions that effectively adjust the variable set towards $\boldsymbol{s}^*$. Specifically, when trajectory reaches $\boldsymbol{s}^*$, the policy $\pi_{(t)}$ is expected to update such that the probability of maintaining this state, $\pi_{(t)}(0|\boldsymbol{s}^*)$, goes to 1 as $t \to \infty$. Conversely, when in states other than $\boldsymbol{s}^*$, the policy should actively seeks actions that bring the state closer to $\boldsymbol{s}^*$, thus progressively improving the model's fit to the data. Specifically, this online computation process is executed using the policy iteration method with Temporal Difference (TD) learning (Sutton & Barto, 2018).

## 3 Policy Iteration with TD Learning

In our variable selection framework, we employ a tailored policy iteration algorithm for progressive refinement of variable adjustment strategies. This approach involves two alternating steps: (1) policy evaluation, and (2) policy improvement. In the policy evaluation step, we estimate the action-value function $Q^{\pi_{\text{old}}}$ for a given policy $\pi_{\text{old}}$, typically by solving the Bellman equation. This step assesses the efficacy of a current policy in terms of how well it selects variables that contribute to an optimal performance in the GLM model. In the policy improvement step, based on the estimated $Q^{\pi_{\text{old}}}$, we derive a new policy $\pi_{\text{new}}$ by choosing actions that maximize this function. This translates to adjusting the variable set by adding or removing variables to enhance the regression model's performance. This iterative process, symbolized as $\pi_{(0)} \xrightarrow{\text{evaluate}} Q^{\pi_{(0)}} \xrightarrow{\text{improve}} \pi_{(1)} \xrightarrow{\text{evaluate}} Q^{\pi_{(1)}} \xrightarrow{\text{improve}} \pi_{(2)} \to \cdots \to \pi^* \to Q^*$, continues until convergence to the optimal policy $\pi^*$ and optimal value $Q^*$.

A key aspect of REVS is balancing exploitation and exploration. Exploitation involves favoring actions that previously resulted in significant rewards, i.e., effectively refining the variable set in the GLM. Conversely, exploration entails trying new actions to potentially uncover better variable combinations, thereby accumulating more substantial long-term rewards. We incorporate this balance by using the $\epsilon$-greedy algorithm (Yang & Zhu, 2002; Chen et al., 2021a) in the policy improvement step, where the optimal action is chosen with probability related to $1 - \epsilon_t$, and other actions are explored with probability related to $\epsilon_t$. The probability distribution for actions in policy $\pi_{(t)}$ at stage $t$ is defined as:

$$\pi_{(t)}(a|\boldsymbol{s}) = \begin{cases} \epsilon_t/(p+1) + 1 - \epsilon_t, & \text{if } a \in \arg \max Q^{\pi_{(t)}}(\boldsymbol{s}, a), \\ \epsilon_t/(p+1), & \text{otherwise.} \end{cases} \quad (5)$$

As the stage progresses ($t \to \infty$), $\epsilon_t$ decreases to 0, leading to the policy towards convergence.

We adapt the TD learning algorithm, specifically the State-Action-Reward-State-Action (SARSA) variant (Zhao et al., 2016; Lee & Kim, 2022; Hu, 2023), to manage the policy iteration process. This on-policy method is particularly suited for variable selection in large state spaces due to its efficiency in learning from incomplete trajectories and updating policies online (Sutton & Barto, 2018). Our SARSA algorithm updates the $Q$-function based on the observed transition from one state-action pair to the next. The update rule is:

$$Q_{t+1}(\boldsymbol{s}_t, a_t) \leftarrow Q_t(\boldsymbol{s}_t, a_t) + \alpha_t \big[ r_t + \gamma Q_t(\boldsymbol{s}_{t+1}, a_{t+1}) - Q_t(\boldsymbol{s}_t, a_t) \big], \quad (6)$$

where $\alpha_t$ is the learning rate that decreases over time, ensuring convergence.

### 3.1 Implementation with Big Data

For the practical implementation of our algorithm, we take linear regression as an illustration example. For any stage $t$, we randomly select *different* $n_{\text{train}}$ observations for the training set and $n_{\text{valid}}$ observations for the validation set from the large dataset, where $n_{\text{train}} \ll n$ and $n_{\text{valid}} \ll n$. We also ensure there's no overlap between the two sets. Let $X_{\boldsymbol{s}_t}$ denote the subset of covariates contained in state $\boldsymbol{s}_t$. Given the training set $\{\boldsymbol{x}_i, y_i\}_{i=1}^{n_{\text{train}}}$, we only use the variables in $X_{\boldsymbol{s}_t}$ to fit a simple linear regression model and obtain the corresponding regression coefficients $\boldsymbol{\zeta}_{\boldsymbol{s}_t}$. Then, the trajectory will transit to $\boldsymbol{s}_{t+1}$ after taking action $a_t$ based on $\pi_{(t)}$. We fit another regression model with $X_{\boldsymbol{s}_{t+1}}$ and obtain the new coefficients $\boldsymbol{\zeta}_{\boldsymbol{s}_{t+1}}$. The change in model performance is calculated by the decline in mean square error $r_{\text{perf},t} = \mathbb{E}_{n_{\text{valid}}}(Y - X_{\boldsymbol{s}_t}^{\intercal}\boldsymbol{\zeta}_{\boldsymbol{s}_t})^2 - \mathbb{E}_{n_{\text{valid}}}(Y - X_{\boldsymbol{s}_{t+1}}^{\intercal}\boldsymbol{\zeta}_{\boldsymbol{s}_{t+1}})^2$, where $\mathbb{E}_{n_{\text{valid}}}$ is the empirical mean with the sampled validation data in stage $t$. The sampling process is implemented at each stage $t$ so that each reward $r_t$ is estimated with different subsets of data. Similarly, in the context of logistic regression, we can follow the same procedure by replacing the decline of mean square error with the increase of log-likelihood function for the logistic model. This maintains the adaptability of our approach across different types of models in GLM. We refer more details for logistic regression in Appendix A.2.

In addressing the challenge of our potentially large state-action space, we recognize that many state-action pairs may not be visited sufficiently, even with the $\epsilon$-greedy approach. This can lead to a prolonged trajectory with numerous stages to gather adequate reward information for updating the $Q$-function, potentially hindering the efficiency of the TD learning algorithm. To mitigate this, we introduce a strategy utilizing quantiles of evolving reward set to guide the trajectory towards more favorable states at each step.

### 3.2 Quantile-guided Navigation for Large Space

The process begins by generating a preliminary trajectory of $T_0$ stages using a uniformly random policy $\pi_{\text{random}}$, where actions for each state are selected with equal probability. During this phase, we record the immediate rewards $r_t^0$ at each stage and compile them into an initial reward set $\mathcal{R}_0 = \{r_t^0\}_{t=1}^{T_0}$. We then define an ascending sequence of quantiles $\{\tau_t\}_{t \geq 0}$, where $\tau_t \to 1$ as $t \to \infty$. As we proceed with TD learning and policy iteration over a main trajectory, capped at a maximum of $T_{\max}$ stages, the reward set is continually updated to $\mathcal{R}_t = \mathcal{R}_{t-1} \bigcup r_t$, incorporating the immediate rewards from each new stage. At each stage $t$, transition to the next state is contingent on the current immediate reward $r_t$ surpassing the upper $\tau_t$-quantile of the evolving reward set $\mathcal{R}_t$. If this condition does not hold, the trajectory is maintained at the current state, exploring alternative actions to transition to other variable sets. This quantile-guided approach guides the trajectory progressively towards better states. By setting $\tau_t \to 1$, the state transitions become increasingly stringent, effectively guiding the trajectory towards better rewarding states as the process evolves.

From Theorem 1 in Section 4, the optimal policy would keep the trajectory stay at the optimal state. So, the final estimation of non-sparse variable set is determined by the states where the action '0' is optimal under the final estimated policy at convergence. This is formalized as $\boldsymbol{s} \in \mathcal{S} : 0 \in \arg\max_{a \in \mathcal{A}} \pi_{\text{final}}(a|\boldsymbol{s})$. Practically, if the state of main trajectory $\boldsymbol{s}_t$ converges to a specific state $\tilde{\boldsymbol{s}}$, it is set to be $\tilde{\boldsymbol{s}}$. The complete algorithm is detailed in Appendix A.

### 3.3 Summarized Advantages of REVS

We summarize the following advantages of our proposed REVS for variable selection in large datasets:

(1) **Computational efficiency.** In scenarios with a large sample size $n$, by computing on small batches of data $(n_{\text{train}}, n_{\text{valid}} \lesssim \log n)$ with MDP rather than solving the optimization problem using the entire dataset, our method reduces the computational cost and hence is more efficient than fitting models on the full dataset. In this way, REVS decomposes the whole variable selection process into a sequence of actions in each stage to refine the variable set. Moreover, even within each small batch, REVS utilizes the quantile-guided navigation technique and focuses on a reduced set of active variables $p_0 \ll p$, which allows for further computational savings compared to other mini-batch methods that operate over all $p$ variables.

(2) **Balance exploration and exploitation.** Traditional variable selection techniques such as backward or forward selection can get trapped in local optima due to their inherently greedy nature, especially in settings with low signal-noise ratio and high-correlated features. By incorporating the $\epsilon$-greedy algorithm, REVS does not only focus on achieving immediate accuracy in variable selection. Instead, it aims to improve long-term accuracy across batches with large datasets, effectively balancing the exploration of new variable sets with the exploitation of known effective ones.

(3) **Prevent overfitting with batch data.** By continual adjusting selected variables with different sampled training and validation batch data, and incorporating cumulative reward information, REVS inherently overcomes overfitting. For high-dimensional setting with correlated features, REVS moderates the influence of individual features that might appear overly predictive in specific subsets, and hence, improving variable selection accuracy.

## 4 Theoretical Analysis

A key aspect in MDP is the use of the Bellman equation from dynamic programming, as highlighted by Sutton & Barto (2018). This recursive equation connects the value of a state to values of its adjacent states, following a specific policy throughout the trajectory. In variable selection, Bellman equations are appropriately adapted to fit this framework. We state the following Bellman equations in our variable selection context. For any given policy $\pi \in \Pi$, we have the Bellman equation for value function:

$$V^\pi(\boldsymbol{s}) = \sum_{a \in \mathcal{A}} \pi(a|\boldsymbol{s}) \left[ R(\boldsymbol{s}, a, \boldsymbol{s} \circ a) + \gamma V^\pi(\boldsymbol{s} \circ a) \right],$$

and the Bellman equation for state-action function:

$$Q^\pi(\boldsymbol{s}, a) = R(\boldsymbol{s}, a, \boldsymbol{s} \circ a) + \gamma \sum_{a' \in \mathcal{A}} \pi(a'|\boldsymbol{s} \circ a) Q^\pi(\boldsymbol{s} \circ a, a'),$$

where $R$ is defined in (3), and $\boldsymbol{s} \circ a$ is denoted as the value of next state $S_{t+1}$ when adjusting variables in current state $S_t = \boldsymbol{s}$ with current action $A_t = a$.

It's worth noting that our Bellman equations, differ from those in standard MDPs. Ours take a simplified version, primarily because the transition probability in our model is a deterministic function dependent on the current state $\boldsymbol{s}$ and action $a$. This means that the value of the subsequent state $\boldsymbol{s} \circ a$ is predetermined once we know $\boldsymbol{s}$ and $a$. Additionally, our model's unique constraint of adding or removing only one variable per stage leads to the value function and $Q$-function in the Bellman equation being specifically relevant to states that differ by only one variable.

Next, we explore how the optimal policy $\pi^*$ in RL guides the progression towards the optimal variable state $\boldsymbol{s}^*$. Traditionally, RL focuses on maximizing discounted cumulative rewards to learn the optimal policy $\pi^*$. In our context, the goal is to identify an optimal adjustment strategy that leads us to the optimal state for variable selection. We aim to establish a connection between these two goals. We use $\mathbb{P}_t^\pi(\boldsymbol{s}', a'|\boldsymbol{s}, a)$ to denote the $t$-step visitation probability $\Pr^\pi(S_t = \boldsymbol{s}', A_t = a'|S_0 = \boldsymbol{s}, A_0 = a)$ on state-action pairs induced by a stationary policy $\pi \in \Pi$. Let $\Delta_a = R(\boldsymbol{s}^*, a, \boldsymbol{s}^* \circ a)$ be the reward for taking a potentially redundant action $a \neq 0$ at the optimal state $\boldsymbol{s}^*$. Intuitively, we expect $\Delta_a$ to be negative, indicating an immediate decrease in reward due to a transition to a less accurate state.

**Assumption 1.** *The immediate reward drop near the optimal state $\boldsymbol{s}^*$ is greater than any future discounted rewards:* $\sup_{\pi \in \Pi} \sum_{t=2}^{\infty} \gamma^t \sum_{\boldsymbol{s}' \in \mathcal{S}, a' \in \mathcal{A}} \mathbb{P}_t^\pi(\boldsymbol{s}', a'|\boldsymbol{s}^*, 0) R(\boldsymbol{s}', a', \boldsymbol{s}' \circ a') < \gamma \inf_{a \in \mathcal{A} \setminus \{0\}} |\Delta_a|$.

For any given $\boldsymbol{s}$ and $a$, $(1-\gamma) \sum_{t=0}^{\infty} \gamma^t \mathbb{P}_t^\pi(\cdot, \cdot|\boldsymbol{s}, a)$ forms a probability mass distribution over state-action pairs. It consists of a mixture of random pairs $\{S_t, A_t\}_{t \geq 0}$ with respective weights $\{(1-\gamma)\gamma^t\}_{t \geq 0}$ starting from $S_0 = \boldsymbol{s}, A_0 = a$. Therefore, $\sum_{t=0}^{\infty} \gamma^t \sum_{\boldsymbol{s}' \in \mathcal{S}, a' \in \mathcal{A}} \mathbb{P}_t^{\overline{\pi}}(\boldsymbol{s}', a'|\boldsymbol{s}, a) R(\boldsymbol{s}', a', \boldsymbol{s}' \circ a')$ can be seen as the expected immediate reward under this transition probability. Assumption 1 implies that the immediate negative impact of deviating from the optimal state $\boldsymbol{s}^*$ is more significant than any future benefits, reinforcing the importance of staying close to $\boldsymbol{s}^*$ for optimal variable selection. Note that Assumption 1 is not trivial, especially when

the model fit does not follow a simple additive relationship when the variables are highly correlated. Then, we present the following theorem for how optimal policy $\pi^*$ guides the state transiting to $\boldsymbol{s}^*$.

**Theorem 1.** *Suppose $\epsilon_t > 0$ for any $t \geq 0$, and $\epsilon_t \to 0$ as $t \to \infty$. Further assume $\sup_{a>0,a\in\mathcal{A}} |f(\boldsymbol{s}^*, a, \boldsymbol{s}^* \circ a)| \leq \lambda \leq \inf_{a<0,a\in\mathcal{A}} |f(\boldsymbol{s}^*, a, \boldsymbol{s}^* \circ a)|$. Let the step size $\alpha_t = \mathcal{O}(t^{-c})$ where $1/2 < c \leq 1$. Then, under Assumption 1, we have (I) $Q_t \to Q^*, \pi_{(t)} \to \pi^*$; (II) Under the optimal policy $\pi^*$, $\pi^*(0|\boldsymbol{s^*}) = 1$, and for any $\boldsymbol{s} \neq \boldsymbol{s}^*$, $\pi^*(0|\boldsymbol{s}) = 0$.*

Theorem 1 reveals that once the state reaches the optimal state $\boldsymbol{s}^*$, it will remain there with no further transitions under the optimal policy $\pi^*$. Essentially, this means that the optimal policy successfully identifies and maintains the ideal variable set. From another perspective, if the current state is not optimal, i.e., either missing important variables or containing redundant ones, the optimal policy will actively work to adjust it, by continually refining the variable set until it aligns with the optimal state $\boldsymbol{s}^*$. This theorem indicates that the optimal policy effectively guides transitions towards achieving the best possible variable selection.

## 5 Experiments

We conduct several experiments to evaluate the performance of our REVS method. We concentrate on linear regression in this section. The results for other GLM types, such as logistic regression, are included in Appendix C.2. For linear regression, we simulate data based on the linear model $Y_i = \beta_0 + \boldsymbol{\beta}^\intercal \boldsymbol{X}_i + \epsilon_i$, where $\epsilon_i \overset{\text{i.i.d}}{\sim} \mathcal{N}(0, 2)$. For logistic regression, we simulate data based on the logistic model $\text{logit}(\mathbb{P}(Y_i = 1)) = \beta_0 + \boldsymbol{\beta}^\intercal \boldsymbol{X}_i$, where $\mathbb{P}(Y_i = 1)$ represents the probability that the binary outcome $Y_i$ is 1, and $\text{logit}(p) = \log\left(\frac{p}{1-p}\right)$ for $p \in (0, 1)$. The predictors $\boldsymbol{X}_i$ are the input features, and $\boldsymbol{\beta}$ represents the coefficients of the model. The dimension of covariates $p$ varies from 200, 400, to 800. We ensure sparsity by setting $p_0 = 25$ for linear regression, where $p_0$ is the count of non-zero true regression coefficients. To characterize the complex correlation structures among $X$, three following structures of the precision matrix $\Omega = \Sigma^{-1}$ for $X$ are considered. See the demonstration of these structures in Figure 2.

**Scenario 1**. ($\Omega$ is block diagonal). True regression coefficients $\boldsymbol{\beta}^* \in \mathbb{R}^p$ are

$$( \underbrace{1, 1, 1, 1, 1, 0, 0, 0, 0, 0}_{\text{Block 1}}, \underbrace{-1, -1, -1, -1, -1, 0, 0, 0, 0, 0}_{\text{Block 2}},$$
$$\underbrace{1, 1, 1, 1, 1, 0, 0, 0, 0, 0}_{\text{Block 3}}, \underbrace{-1, -1, -1, -1, -1, 0, 0, 0, 0, 0}_{\text{Block 4}},$$
$$\underbrace{1, 1, 1, 1, 1, 0, 0, 0, 0, 0}_{\text{Block 5}}, 0, 0, \ldots, 0)^\intercal.$$

We generate covariates with $X_{5k+j} = Z_k + \epsilon^x_{5k+j}$ where $Z_k \overset{\text{i.i.d}}{\sim} \mathcal{N}(0, 1)$, for $0 \leq k \leq 4$ and $1 \leq j \leq 5$, and $\epsilon^x_j \overset{\text{i.i.d}}{\sim} \mathcal{N}(0, 1)$ for $1 \leq j \leq 50$. Remaining $X_j \overset{\text{i.i.d}}{\sim} \mathcal{N}(0, 1)$ for $51 \leq j \leq p$.

**Scenario 2**. ($\Omega$ is banded). The setup for $\boldsymbol{\beta}^* \in \mathbb{R}^p$ is identical to Scenario 1. The covarates are generated with $X \sim \mathcal{N}(\boldsymbol{0}, \Sigma)$ where $\Sigma_{ij} = 0.8^{|i-j|}$.

**Scenario 3**. ($\Omega$ is sparse). $\boldsymbol{\beta}^* \in \mathbb{R}^p$ are the same as Scenario 1. The covarates are generated with $X \sim \mathcal{N}(\boldsymbol{0}, \Omega^{-1})$ where $\Omega = B + \delta I$. Here, $I$ is an identity matrix, and $B$ has off-diagonal entries, which equal to 0.5 with probability 0.6, and 0 with probability 0.4.

We compare our proposed method with other state-of-art penalized regression methods for variable selection. The following five methods are compared: (1) Lasso (Tibshirani, 1996); (2) Elastic net (Zou & Hastie, 2005); (3) Smoothly Clipped Absolute Deviation (SCAD) based penalty (Fan & Li, 2001); (4) Minimax Concave Penalty (MCP) (Zhang, 2010); (5) Our proposed REVS method with TD-learning.

We categorize variables with non-zero coefficients using positive labels, while those with zero coefficients are assigned negative labels. Then, we employ the following criteria to assess the performance of comparison methods: (1) Number of variables selected in the final model, with true value $p_0 = 25$; (2) True Negative Rate

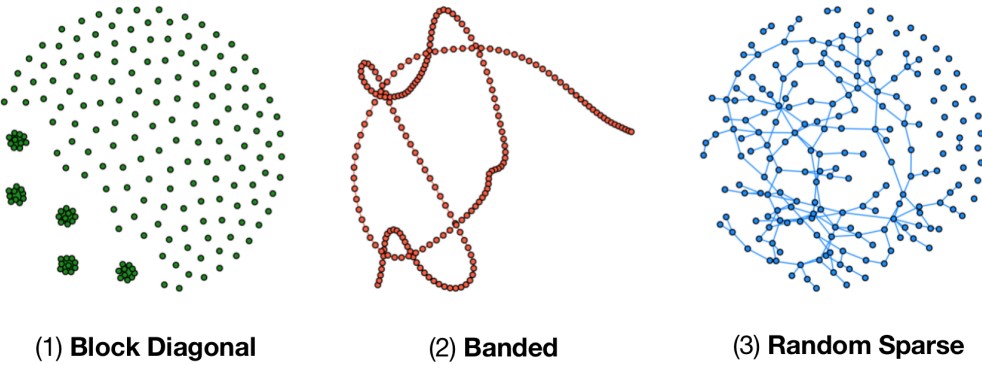

(1) **Block Diagonal**    (2) **Banded**    (3) **Random Sparse**

Figure 2: Three structures of precision matrix.

(TNR) for selected variables; (3) Positive Predictive Value (PPV) for selected variables; (4) Mean Square Error (MSE) for predicting $Y$ on testing data.

## 5.1 Big Data Setting

We set the number of observations $n = 500,000$. A dataset of size $n_{\text{eval}} = 5,000$ is generated to assess model performance. For big data settings, REVS streamlines the iterative process by sampling a small number of observations, setting both $n_{\text{train}} = n_{\text{test}} = 200$. We set preliminary trajectory length $T_0 = 250$. The main trajectory extends up to $T_{\max} = 2000$ stages, with the quantile sequence $\tau_t = 0.6 + 0.4t/T_{\max} \to 1$ as $t \to T_{\max}$. We use a discount factor $\gamma = 0.99$ and a penalty term parameter $\lambda = 0.2$. To ensure convergence of TD learning, we set the learning rate $\alpha_t = 1/t$ and exploration probability $\epsilon_t = 1/t^{0.1}$, respectively. To enhance computational efficiency in the big data scenario, we adopt the divide and conquer strategy (Smith, 1985) for penalized models. We split the data into 400 folds and aggregate the selected variables, by employing a frequency table with a threshold cutting ratio of 0.8. The simulations for each scenario is replicated 50 times.

In our analysis, all methods exhibited similar results in terms of TNR, PPV, and MSE for variable selection accuracy, largely attributed to the large sample sizes (see Figure 5 in Appendix C.1). However, the REVS method distinguished itself by demonstrating superior computational efficiency. This advantage is due to the method's reliance on fitting simple linear models with smaller sample sizes at each stage. In Table 1, we underscore this computational efficiency by presenting the mean computation time (in minutes) across 50 replications for linear regression, demonstrating REVS's effectiveness in handling big data.

## 5.2 High-dimensional Setting

We explore our method in high-dimensional settings, where the sample size $n = 200$ with $n_{\text{train}} = n_{\text{test}} = 150$. The other parameter selection remain consistent with those in the big data setting. Figure 3(c) and Table 2 demonstrate that REVS not only closely aligns with the optimal variable selection, with the number of variables selected being nearest to the true count $p_0 = 25$, but also exhibits the lowest variability for linear regression. Furthermore, Figures 3(a)(b), and 4 highlight REVS's superior performance in terms of TNR, PPV, and MSE respectively. This is significant as it indicates a lower likelihood of REVS selecting redundant variables compared to other penalized variable selection techniques. In addition to its strong performance in linear regression, REVS also yields impressive results for logistic regression, as demonstrated in Table 3 and Figure 6. This suggests that REVS effectively avoids redundant variable selection, maintaining high accuracy and stability in both linear and logistic regression scenarios.

By iteratively updating selected variables using sampled training and validation subsets, we effectively harness batch data to calculate the reward $r_t$. This approach is able to reduce the risk of overfitting and inclusion of redundant variables, and foster more informed model selection at each stage $t$ through the integration of

Table 1: Means of computation time (minutes) in *big data setting* under 50 replications. Best values in bold.

| Structure | Method | $p = 200$ | $p = 400$ | $p = 800$ |
|-----------|--------|-----------|-----------|-----------|
| Block $\Omega$ | Lasso | 1.62 | 5.47 | 13.99 |
| | Elastic | 7.20 | 34.25 | 43.80 |
| | SCAD | 1.69 | 3.07 | 6.28 |
| | MCP | 1.65 | 2.96 | 6.34 |
| | REVS | **0.34** | **0.82** | **3.38** |
| Banded $\Omega$ | Lasso | 3.38 | 6.21 | 8.09 |
| | Elastic | 9.80 | 16.18 | 25.88 |
| | SCAD | 7.44 | 8.10 | 17.31 |
| | MCP | 8.57 | 9.33 | 20.12 |
| | REVS | **0.60** | **1.82** | **2.78** |
| Sparse $\Omega$ | Lasso | 2.45 | 3.51 | 17.54 |
| | Elastic | 7.27 | 10.28 | 54.96 |
| | SCAD | 2.38 | 3.08 | 8.31 |
| | MCP | 2.28 | 2.30 | 9.03 |
| | REVS | **0.40** | **0.81** | **4.04** |

cumulative reward insights. This feature of REVS underscores its robustness and adaptability, effectively extending its range of applicability from large-scale data scenarios to more challenging high-dimensional settings.

| | $p = 200$ | | | $p = 400$ | | | $p = 800$ | | |
|---|---|---|---|---|---|---|---|---|---|
| | # | TNR(%) | PPV(%) | # | TNR(%) | PPV(%) | # | TNR(%) | PPV(%) |
| Block Structure | | | | | | | | | |
| Lasso | 77.9 | 69.8% | 32.9% | 102.4 | 79.3% | 25.2% | 126.6 | 86.9% | 20.3% |
| Elastic | 81.0 | 68.0% | 31.7% | 104.7 | 78.8% | 25.4% | 127.1 | 86.8% | 20.3% |
| SCAD | 43.9 | 89.2% | 57.6% | 55.3 | 91.9% | 46.1% | 69.2 | 94.3% | 36.5% |
| MCP | 33.4 | 95.2% | 75.9% | 35.8 | 97.1% | 71.0% | 41.2 | 97.9% | 61.4% |
| REVS | **26.1** | **99.4%** | **96.0%** | **26.4** | **99.4%** | **94.6%** | **28.5** | **99.5%** | **86.5%** |
| Banded Structure | | | | | | | | | |
| Lasso | 56.6 | 82.0% | 46.2% | 61.5 | 90.2% | 44.8% | 75.0 | 93.5% | 36.5% |
| Elastic | 54.4 | 83.2% | 48.4% | 60.0 | 90.8% | 43.0% | 78.4 | 93.1% | 33.0% |
| SCAD | 26.9 | 98.8% | 98.7% | 27.3 | 99.4% | 97.5% | 28.2 | 99.6% | 95.6% |
| MCP | 26.8 | 99.1% | 99.2% | 27.6 | 99.6% | 98.8% | 28.7 | 99.5% | 98.3% |
| REVS | **25.0** | **100%** | **100%** | **25.0** | **100%** | **100%** | **25.0** | **100%** | **100%** |
| Sparse Structure | | | | | | | | | |
| Lasso | 79.8 | 68.7% | 32.3% | 101.0 | 79.7% | 25.6% | 129.8 | 86.5% | 19.6% |
| Elastic | 84.9 | 65.8% | 30.0% | 108.0 | 77.9% | 24.1% | 126.5 | 86.9% | 20.6% |
| SCAD | 29.2 | 97.6% | 88.3% | 30.6 | 98.5% | 90.6% | 31.2 | 99.2% | 94.5% |
| MCP | 26.7 | 99.3% | 97.1% | 26.9 | 99.5% | 96.0% | 27.2 | 99.7% | 89.9% |
| REVS | **25.6** | **99.6%** | **97.7%** | **26.1** | **99.7%** | **97.2%** | **26.5** | **99.8%** | **95.8%** |

Table 2: Means of number of selected variables (#) with $p_0 = 25$, Positive Predictive Value (PPV), and True Negative Rate (TNR) in *high-dimensional setting* for *linear regression* under 50 replications. Best values in bold.

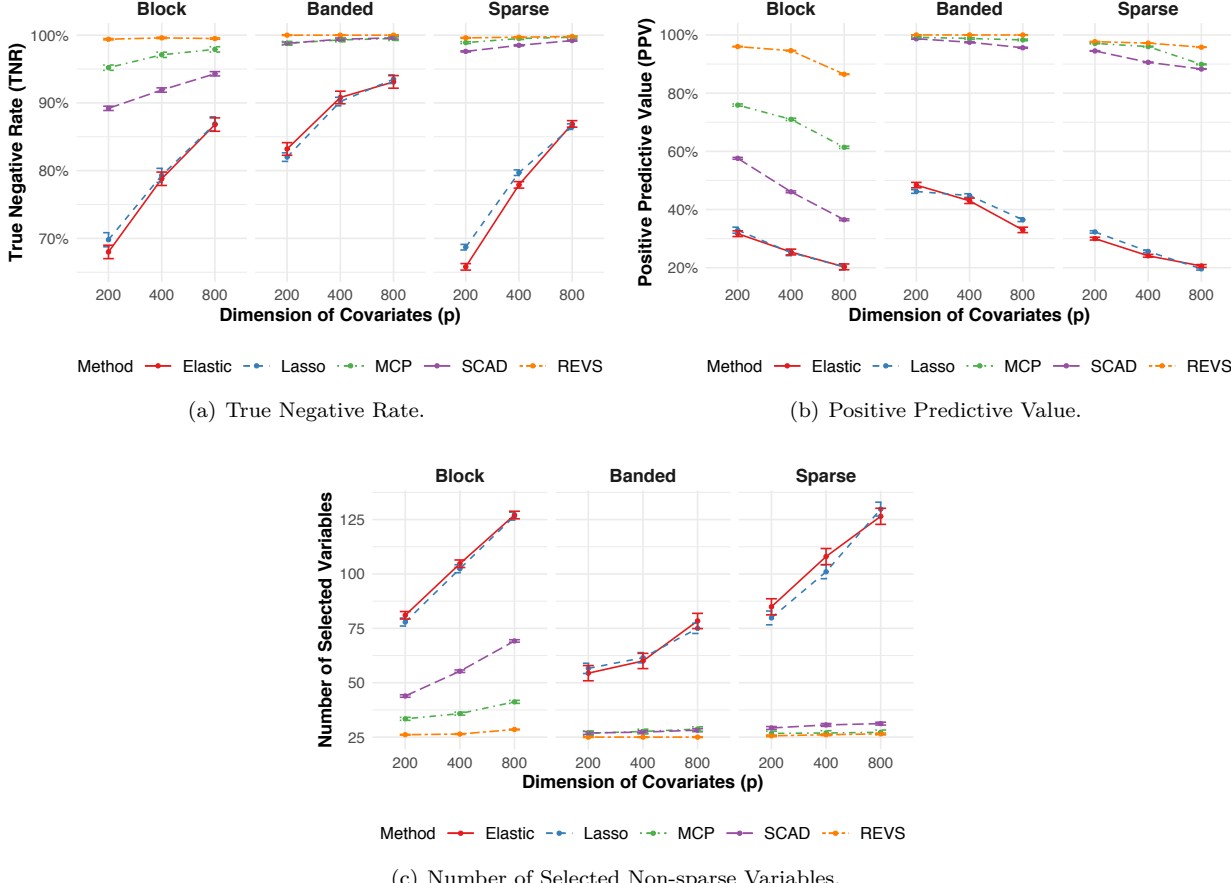

(a) True Negative Rate.

(b) Positive Predictive Value.

(c) Number of Selected Non-sparse Variables.

Figure 3: True Negative Rates (TNR), Positive Predictive Values (PPV), and numbers of selected non-sparse variables of comparison methods in *linear regression* with *high-dimensional data setting*. The statistics in the text show the mean and variance of above metrics across 50 replications.

## 6 Conclusions

In this paper, we adapted RL to address key challenges in supervised learning, particularly focusing on feature selection in big data. We formulate the problem as a *time-homogeneous MDP* and develop an efficient policy-iteration method using the TD learning algorithm to solve it. Our proposed REVS dynamically adjusts the selection of variables in a step-wise manner, utilizing different sampled training and validation subsets. By integrating the $\epsilon$-greedy algorithm, REVS transcends beyond achieving immediate accuracy in variable selection. It is designed to enhance long-term accuracy across large dataset batches, adeptly balancing the exploration of new variable combinations with the exploitation of established effective ones. In simulation studies, REVS shows significant computational efficiency for handling big data, surpassing other state-of-the-art variable selection methods reliant on penalized regression. Furthermore, the utilization of batch data for cumulative reward computation inherently safeguards against overfitting and continually guides the variable selection process. REVS has also demonstrated superior performance in variable selection accuracy, particularly in high-dimensional datasets with highly correlated covariates.

For future extensions, REVS currently utilizes the $\epsilon$-greedy algorithm. The exploration of other RL algorithms, such as the Upper Confidence Bound (UCB) (Garivier & Moulines, 2011; Kaufmann et al., 2012) and Thompson Sampling (Kang et al., 2024), presents another exciting research prospect. We leave these interesting directions for future research.

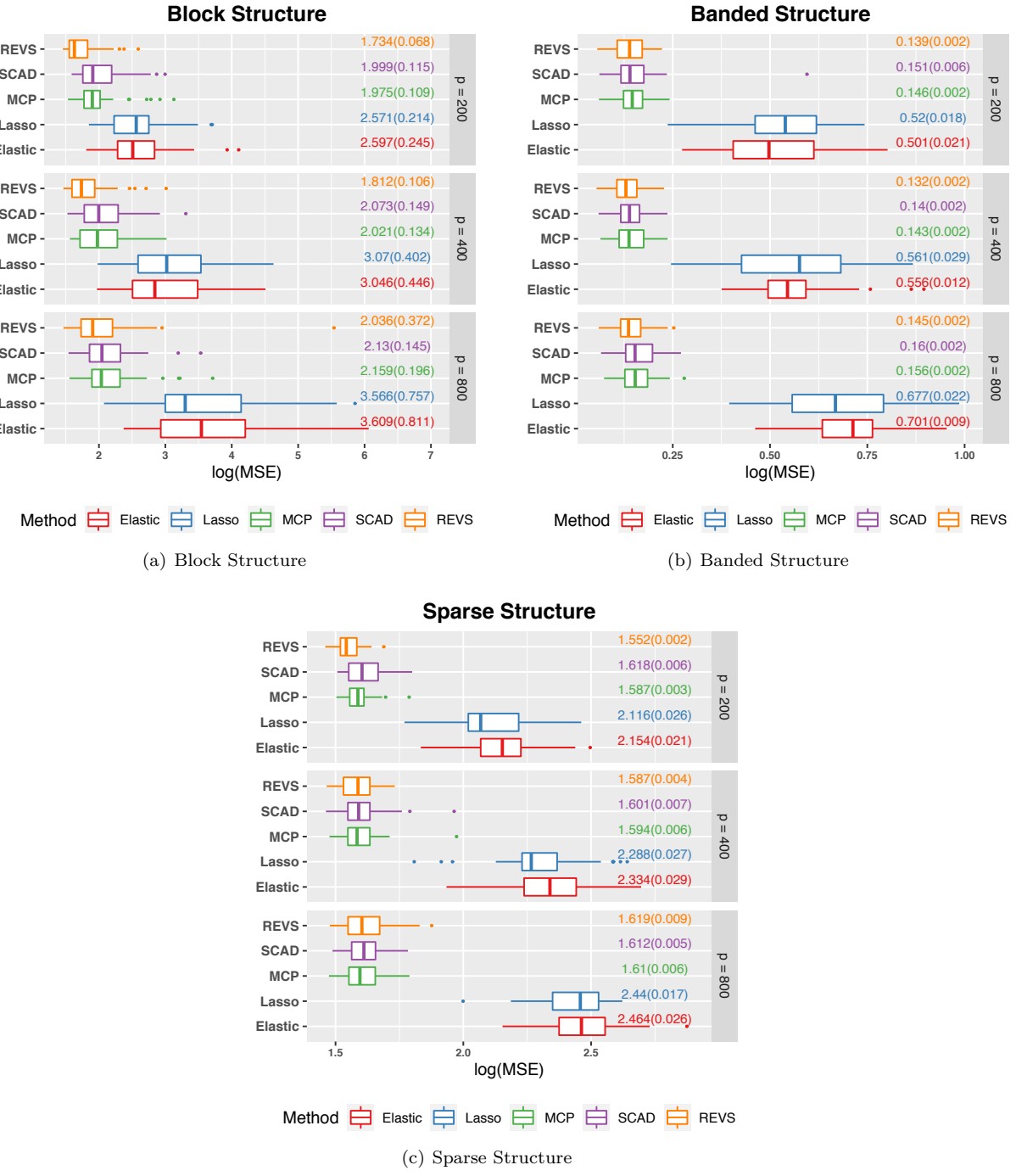

(a) Block Structure

(b) Banded Structure

(c) Sparse Structure

Figure 4: Mean Square Error (MSE) for predicting $Y$ on testing data in *linear regression* with *high-dimensional data setting*. The statistics in the text show the mean and variance of MSE across 50 replications.

## Broader Impact Statement

The goal of this paper is to advance the problem of variable selection in the field of Machine Learning. We believe that advancements in this area have the great potential to create widespread societal implications. While our work opens up possibilities for positive change, it is beyond the scope of this paper to delve into the specifics of these societal impacts.

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

# A Appendix A: Additional Implementation Details

## A.1 Detailed Algorithm for REVS in Linear Regression

---

**Algorithm 1** REVS for Linear Regression

---

1. **Initialize** $Q_0$, $\{\tau_t\}_{t \geq 0}$.
2. **Generate** *preliminary* trajectory with *random* policy:
    **Initialize** $s_0$;
    **Sample** $a_0$ from *random* policy $\pi_{\mathrm{random}}(\cdot|s_0)$;
    **For** stage $t = 0, 1, 2, \ldots, T_0$, **do**:
        **Take** action $a_t$ and **transit** to $s_{t+1}$ by (2);
        **Receive** reward $r_t^0$ defined in (3):
            **Sample** training set $\{x_i, y_i\}_{i=1}^{n_{\mathrm{train}}}$ and validation set $\{x_j, y_j\}_{j=1}^{n_{\mathrm{valid}}}$ from whole data;
            **Estimate** $\zeta_{s_t}$ and $\zeta_{s_{t+1}}$ by fitting $Y \sim X_{s_t}$ and $Y \sim X_{s_{t+1}}$ with $\{x_i, y_i\}_{i=1}^{n_{\mathrm{train}}}$;
            **Set** $r_{\mathrm{perf},t}^0 = \mathbb{E}_{n_{\mathrm{valid}}}(Y - X_{s_t}^{\intercal}\zeta_{s_t})^2 - \mathbb{E}_{n_{\mathrm{valid}}}(Y - X_{s_{t+1}}^{\intercal}\zeta_{s_{t+1}})^2$;
            **Set** $r_{\mathrm{pen},t}^0 \leftarrow -\lambda\mathbb{I}[a_t \in \{+1, +2, \ldots, +p\}] + \lambda\mathbb{I}[a_t \in \{-1, -2, \ldots, -p\}]$;
            **Obtain** $r_t^0 \leftarrow r_{\mathrm{perf},t}^0 + r_{\mathrm{pen},t}^0$.
        **Sample** $a_{t+1}$ from $\pi_{\mathrm{random}}(\cdot|s_{t+1})$;
        **Update** $Q_{t+1}(s_t, a_t) \leftarrow Q_t(s_t, a_t) + \alpha_t[r_t + \gamma Q_t(s_{t+1}, a_{t+1}) - Q_t(s_t, a_t)]$;
        **Update** $s_t \leftarrow s_{t+1}, a_t \leftarrow a_{t+1}$, store $r_t^0$;
    **Return** $\mathcal{R}_0 = \{r_t^0\}_{t=1}^{T_0}$ and $Q_{T_0}$.
3. **Generate** *main* trajectory with $\epsilon$-*greedy* policy:
    **Initialize** $s_0$, and **set** $Q_0 \leftarrow Q_{T_0}, \mathcal{R}_0 \leftarrow \mathcal{R}_0$ obtained from previous step;
    **Sample** $a_0$ from $\epsilon$-*greedy* policy $\pi_{(0)}(\cdot|s_0)$ based on $Q_0$ by (5);
    **For** stage $t = 0, 1, 2, \ldots, T_{\max}$, **do**:
        **Take** action $a_t$ and **transit** to $s_{t+1}$ by (2);
        **Receive** reward $r_t \leftarrow r_{\mathrm{perf},t} + r_{\mathrm{pen},t}$ in (3) with batch data as in *preliminary* trajectory;
        **Sample** $a_{t+1}$ from $\epsilon$-*greedy* policy $\pi_{(t)}(\cdot|s_{t+1})$ by (5) based on $Q_t$;
        **Update** $Q_{t+1}(s_t, a_t) \leftarrow Q_t(s_t, a_t) + \alpha_t[r_t + \gamma Q_t(s_{t+1}, a_{t+1}) - Q_t(s_t, a_t)]$;
        **If** $r_t > \mathrm{quantile}(\mathcal{R}_t, \tau_t)$:
            **Update** $s_t \leftarrow s_{t+1}, a_t \leftarrow a_{t+1}$;
        **Update** $\mathcal{R}_{t+1} \leftarrow \mathcal{R}_t \bigcup r_t$:
    **Until** trajectory converges to $\tilde{s}$.
4. **Set** final variable set to be $\tilde{s}$ if $s_t \to \tilde{s}$, otherwise sample a variable set from $\{s \in \mathcal{S} : 0 \in \arg\max_{a \in \mathcal{A}} \pi_{(T_{\max})}(a|s)\}$.

---

## A.2 REVS for Logistic Regression

In a manner akin to linear regression, for each stage $t$, we select a subset of observations from the larger dataset for training ($n_{\text{train}}$) and validation ($n_{\text{valid}}$), where both $n_{\text{train}}$ and $n_{\text{valid}}$ are significantly smaller than the total number of observations $n$. Let $X_{\boldsymbol{s}_t}$ denote the subset of covariates contained in state $\boldsymbol{s}_t$. Given the training set $\{\boldsymbol{x}_i, y_i\}_{i=1}^{n_{\text{train}}}$, we fit a logistic regression model using only the variables in $X_{\boldsymbol{s}_t}$ and obtain the regression coefficients $\boldsymbol{\eta}_{\boldsymbol{s}_t}$. After taking action $a_t$ according to policy $\pi_{(t)}$, the state transitions to $\boldsymbol{s}_{t+1}$, where we fit a new logistic regression model with $X_{\boldsymbol{s}_{t+1}}$ and obtain coefficients $\boldsymbol{\eta}_{\boldsymbol{s}_{t+1}}$.

In the logistic regression framework, we modify the model performance metric. Instead of tracking the decline in mean square error as in linear regression, we focus on the increase in the log-likelihood function. The change in model performance, denoted as $r_{\text{perf},t}$, is defined by the increase in log-likelihood:

$$r_{\text{perf},t} = \mathbb{E}_{n_{\text{valid}}}[Y X_{\boldsymbol{s}_{t+1}}^{\mathsf{T}} \boldsymbol{\eta}_{\boldsymbol{s}_{t+1}} - \log(1 + \exp(X_{\boldsymbol{s}_{t+1}}^{\mathsf{T}} \boldsymbol{\eta}_{\boldsymbol{s}_{t+1}}))] - \mathbb{E}_{n_{\text{valid}}}[Y X_{\boldsymbol{s}_t}^{\mathsf{T}} \boldsymbol{\eta}_{\boldsymbol{s}_t} - \log(1 + \exp(X_{\boldsymbol{s}_t}^{\mathsf{T}} \boldsymbol{\eta}_{\boldsymbol{s}_t}))].$$

Here, $\mathbb{E}_{n_{\text{valid}}}$ represents the empirical mean calculated using the validation data sampled at stage $t$. This shift in performance metric from MSE to log-likelihood is a key aspect of adapting REVS to logistic regression.

---

**Algorithm 2** REVS for Logistic Regression

---

1. **Initialize** $Q_0, \{\tau_t\}_{t \geq 0}$.
2. **Generate** *preliminary* trajectory with *random* policy:
    **Initialize** $\boldsymbol{s}_0$;
    **Sample** $a_0$ from *random* policy $\pi_{\text{random}}(\cdot|\boldsymbol{s}_0)$;
    **For** stage $t = 0, 1, 2, \ldots, T_0$, **do**:
        **Take** action $a_t$ and **transit** to $\boldsymbol{s}_{t+1}$ by (2);
        **Receive** reward $r_t^0$ defined in (3):
            **Sample** training set $\{\boldsymbol{x}_i, y_i\}_{i=1}^{n_{\text{train}}}$ and validation set $\{\boldsymbol{x}_j, y_j\}_{j=1}^{n_{\text{valid}}}$ from whole data;
            **Estimate** $\boldsymbol{\eta}_{\boldsymbol{s}_t}$ and $\boldsymbol{\eta}_{\boldsymbol{s}_{t+1}}$ by fitting $Y \sim X_{\boldsymbol{s}_t}$ and $Y \sim X_{\boldsymbol{s}_{t+1}}$ with $\{\boldsymbol{x}_i, y_i\}_{i=1}^{n_{\text{train}}}$;
            **Set** $r_{\text{perf},t}^0 = \mathbb{E}_{n_{\text{valid}}}[Y X_{\boldsymbol{s}_{t+1}}^{\mathsf{T}} \boldsymbol{\eta}_{\boldsymbol{s}_{t+1}} - \log(1 + \exp(X_{\boldsymbol{s}_{t+1}}^{\mathsf{T}} \boldsymbol{\eta}_{\boldsymbol{s}_{t+1}}))] - \mathbb{E}_{n_{\text{valid}}}[Y X_{\boldsymbol{s}_t}^{\mathsf{T}} \boldsymbol{\eta}_{\boldsymbol{s}_t} - \log(1 + \exp(X_{\boldsymbol{s}_t}^{\mathsf{T}} \boldsymbol{\eta}_{\boldsymbol{s}_t}))]$;
            **Set** $r_{\text{pen},t}^0 \leftarrow -\lambda \mathbb{I}[a_t \in \{+1, +2, \ldots, +p\}] + \lambda \mathbb{I}[a_t \in \{-1, -2, \ldots, -p\}]$;
            **Obtain** $r_t^0 \leftarrow r_{\text{perf},t}^0 + r_{\text{pen},t}^0$.
        **Sample** $a_{t+1}$ from $\pi_{\text{random}}(\cdot|\boldsymbol{s}_{t+1})$;
        **Update** $Q_{t+1}(\boldsymbol{s}_t, a_t) \leftarrow Q_t(\boldsymbol{s}_t, a_t) + \alpha_t[r_t + \gamma Q_t(\boldsymbol{s}_{t+1}, a_{t+1}) - Q_t(\boldsymbol{s}_t, a_t)]$;
        **Update** $\boldsymbol{s}_t \leftarrow \boldsymbol{s}_{t+1}, a_t \leftarrow a_{t+1}$, store $r_t^0$;
    **Return** $\mathcal{R}_0 = \{r_t^0\}_{t=1}^{T_0}$ and $Q_{T_0}$.
3. **Generate** *main* trajectory with $\epsilon$*-greedy* policy:
    **Initialize** $\boldsymbol{s}_0$, and **set** $Q_0 \leftarrow Q_{T_0}, \mathcal{R}_0 \leftarrow \mathcal{R}_0$ obtained from previous step;
    **Sample** $a_0$ from $\epsilon$*-greedy* policy $\pi_{(0)}(\cdot|\boldsymbol{s}_0)$ based on $Q_0$ by (5);
    **For** stage $t = 0, 1, 2, \ldots, T_{\max}$, **do**:
        **Take** action $a_t$ and **transit** to $\boldsymbol{s}_{t+1}$ by (2);
        **Receive** reward $r_t \leftarrow r_{\text{perf},t} + r_{\text{pen},t}$ in (3) with batch data as in *preliminary* trajectory;
        **Sample** $a_{t+1}$ from $\epsilon$*-greedy* policy $\pi_{(t)}(\cdot|\boldsymbol{s}_{t+1})$ by (5) based on $Q_t$;
        **Update** $Q_{t+1}(\boldsymbol{s}_t, a_t) \leftarrow Q_t(\boldsymbol{s}_t, a_t) + \alpha_t[r_t + \gamma Q_t(\boldsymbol{s}_{t+1}, a_{t+1}) - Q_t(\boldsymbol{s}_t, a_t)]$;
        **If** $r_t > \text{quantile}(\mathcal{R}_t, \tau_t)$:
            **Update** $\boldsymbol{s}_t \leftarrow \boldsymbol{s}_{t+1}, a_t \leftarrow a_{t+1}$;
        **Update** $\mathcal{R}_{t+1} \leftarrow \mathcal{R}_t \bigcup r_t$:
    **Until** trajectory converges to $\tilde{\boldsymbol{s}}$.
4. **Set** final variable set to be $\tilde{\boldsymbol{s}}$ if $\boldsymbol{s}_t \to \tilde{\boldsymbol{s}}$, otherwise sample a variable set from $\{\boldsymbol{s} \in \mathcal{S} : 0 \in \arg\max_{a \in \mathcal{A}} \pi_{(T_{\max})}(a|\boldsymbol{s})\}$.

---

# B    Appendix B: Proof of Theoretical Results

## B.1    Proof of Bellman Equation under Variable Selection Context

The value function $V(s)$ represents the expected reward when starting in state $s$ and following a certain policy $\pi$ thereafter. The Bellman equation for the value function in a general RF problem is:

$$V^{\pi}(s) = \sum_{a \in \mathcal{A}} \pi(a|s) \sum_{s' \in \mathcal{S}} \mathbb{P}(s'|s, a) \left[ R(s, a, s') + \gamma V^{\pi}(s') \right].$$

Note that in our defined MDP, the transition probability is a deterministic function given $s$ and $a$. Hence,

$$V^{\pi}(s) = \sum_{a \in \mathcal{A}} \pi(a|s) \sum_{s' \in \mathcal{S}} \mathbb{I}(s' = s \circ a) \left[ R(s, a, s') + \gamma V^{\pi}(s') \right]$$
$$= \sum_{a \in \mathcal{A}} \pi(a|s) \left[ R(s, a, s \circ a) + \gamma V^{\pi}(s \circ a) \right].$$

Similarly, the action-value function $Q(s, a)$ represents the expected return after taking an action $a$ in state $s$ and then following policy $\pi$. From the general Bellman equation for the $Q$-function, we have

$$Q^{\pi}(s, a) = \sum_{s' \in \mathcal{S}} \mathbb{P}(s'|s, a) \left[ R(s, a, s') + \gamma \sum_{a' \in A} \pi(a'|s') Q^{\pi}(s', a') \right]$$
$$= \sum_{s' \in \mathcal{S}} \mathbb{I}(s' = s \circ a) \left[ R(s, a, s') + \gamma \sum_{a' \in A} \pi(a'|s') Q^{\pi}(s', a') \right]$$
$$= R(s, a, s \circ a) + \gamma \sum_{a' \in A} \pi(a'|s \circ a) Q^{\pi}(s \circ a, a').$$

## B.2    Useful Lemma for Proof of Theorem 1

We restate the following lemma (Corollary 1.5 and Lemma 1.6 in Agarwal et al. (2019)) for proof of Theorem 1. Let $\mathbb{P}^{\pi}$ to be the transition matrix on state-action pairs induced by a stationary policy $\pi$. Specifically, $\mathbb{P}^{\pi}_{(s,a),(s',a')} := \mathbb{P}(s' \mid s, a)\pi(a' \mid s')$. Then we have that:

$$[(1-\gamma)(I - \gamma \mathbb{P}^{\pi})^{-1}]_{(s,a),(s',a')} = (1-\gamma) \sum_{t=0}^{\infty} \gamma^t \mathbb{P}^{\pi}(s_t = s', a_t = a' \mid s_0 = s, a_0 = a),$$

and

$$Q^{\pi} = (I - \gamma \mathbb{P}^{\pi})^{-1} R.$$

## B.3    Proof of Theorem 1

For (I), note that our formulation defines a valid MDP, and our choices of $\alpha_t$ and $\epsilon_t$ satisfy the Robbins-Monro conditions (Singh et al., 2000). Then, based on Theorem 1 in Singh et al. (2000), we have $Q_t \to Q^*$ and $\pi_{(t)} \to \pi^*$.

For (II), we firstly prove that, under the optimal policy $\pi^*$, we have $\pi^*(0|s^*) = 1$. We start from the Q-function. Suppose we have a policy $\pi_0$ where $\pi_0(0|s^*) = 1$. Then, from Bellman equation, we have

$$Q^{\pi_0}(s^*, 0) = R(s^*, 0, s^* \circ 0) + \gamma \sum_{a' \in \mathcal{A}} \pi_0(a'|s^* \circ 0) Q^{\pi_0}(s^* \circ 0, a')$$
$$= R(s^*, 0, s^*) + \gamma \sum_{a' \in \mathcal{A}} \pi_0(a'|s^*) Q^{\pi}(s^*, a')$$
$$= 0 + \gamma Q^{\pi_0}(s^*, 0).$$

So, we get $Q^{\pi_0}(\boldsymbol{s}^*, 0) = 0$. Based on the definition of the optimal policy $\pi^*$, $Q^{\pi^*}(\boldsymbol{s}^*, 0) \geq Q^{\pi_0}(\boldsymbol{s}^*, 0) = 0$.

Now, we consider any policy $\pi_1$ where $\pi_1(a \neq 0|\boldsymbol{s}^*) > 0$. In other words, $\pi_1$ tends to adjust the variables when the trajectory reaches the optimal states. From above Lemma in Appendix B.2, we can obtain a close form to characterize the $Q$-function with Bellman Equation. In particular,

$$Q^{\pi_1}(\boldsymbol{s}^*, 0) = \sum_{\boldsymbol{s}' \in \mathcal{S}} \sum_{a' \in \mathcal{A}} \sum_{t=0}^{\infty} \gamma^t \mathbb{P}_t^{\pi_1}(\boldsymbol{s}', a'|\boldsymbol{s}^*, 0) R(\boldsymbol{s}', a', \boldsymbol{s}' \circ a')$$

$$= 0 + \sum_{\boldsymbol{s}' \in \mathcal{S}} \sum_{a' \in \mathcal{A} \setminus \{0\}} \sum_{t=0}^{\infty} \gamma^t \mathbb{P}_t^{\pi_1}(\boldsymbol{s}', a'|\boldsymbol{s}^*, 0) R(\boldsymbol{s}', a', \boldsymbol{s}' \circ a')$$

$$= 0 + \mathbb{I}(\boldsymbol{s}' = \boldsymbol{s}^*, a' = 0) R(\boldsymbol{s}^*, 0, \boldsymbol{s}^* \circ 0) + \gamma \sum_{\boldsymbol{s}' \in \mathcal{S}} \sum_{a' \in \mathcal{A} \setminus \{0\}} \mathbb{P}_1^{\pi_1}(\boldsymbol{s}', a'|\boldsymbol{s}^*, 0) R(\boldsymbol{s}', a', \boldsymbol{s}' \circ a')$$

$$+ \sum_{\boldsymbol{s}' \in \mathcal{S}} \sum_{a' \in \mathcal{A} \setminus \{0\}} \sum_{t=2}^{\infty} \gamma^t \mathbb{P}_t^{\pi_1}(\boldsymbol{s}', a'|\boldsymbol{s}^*, 0) R(\boldsymbol{s}', a', \boldsymbol{s}' \circ a')$$

$$= 0 + 0 + \gamma \sum_{\boldsymbol{s}' \in \mathcal{S}} \sum_{a' \in \mathcal{A} \setminus \{0\}} \mathbb{P}(\boldsymbol{s}'|\boldsymbol{s}^*, 0) \pi_1(a'|\boldsymbol{s}') R(\boldsymbol{s}', a', \boldsymbol{s}' \circ a')$$

$$+ \sum_{\boldsymbol{s}' \in \mathcal{S}} \sum_{a' \in \mathcal{A} \setminus \{0\}} \sum_{t=2}^{\infty} \gamma^t \mathbb{P}_t^{\pi_1}(\boldsymbol{s}', a'|\boldsymbol{s}^*, 0) R(\boldsymbol{s}', a', \boldsymbol{s}' \circ a')$$

$$= 0 + \gamma \sum_{a' \in \mathcal{A} \setminus \{0\}} \pi_1(a'|\boldsymbol{s}^*) R(\boldsymbol{s}^*, a', \boldsymbol{s}^* \circ a')$$

$$+ \sum_{\boldsymbol{s}' \in \mathcal{S}} \sum_{a' \in \mathcal{A} \setminus \{0\}} \sum_{t=2}^{\infty} \gamma^t \mathbb{P}_t^{\pi_1}(\boldsymbol{s}', a'|\boldsymbol{s}^*, 0) R(\boldsymbol{s}', a', \boldsymbol{s}' \circ a')$$

$$= \gamma \sum_{a \in \mathcal{A} \setminus \{0\}} \pi_1(a|\boldsymbol{s}^*) \Delta_a + \sum_{\boldsymbol{s}' \in \mathcal{S}} \sum_{a' \in \mathcal{A} \setminus \{0\}} \sum_{t=2}^{\infty} \gamma^t \mathbb{P}_t^{\pi_1}(\boldsymbol{s}', a'|\boldsymbol{s}^*, 0) R(\boldsymbol{s}', a', \boldsymbol{s}' \circ a')$$

$$\leq \gamma \sup_{a \in \mathcal{A} \setminus \{0\}} \Delta_a + \sum_{\boldsymbol{s}' \in \mathcal{S}} \sum_{a' \in \mathcal{A} \setminus \{0\}} \sum_{t=2}^{\infty} \gamma^t \mathbb{P}_t^{\pi_1}(\boldsymbol{s}', a'|\boldsymbol{s}^*, 0) R(\boldsymbol{s}', a', \boldsymbol{s}' \circ a')$$

$$= -\gamma \inf_{a \in \mathcal{A} \setminus \{0\}} |\Delta_a| + \sum_{\boldsymbol{s}' \in \mathcal{S}} \sum_{a' \in \mathcal{A} \setminus \{0\}} \sum_{t=2}^{\infty} \gamma^t \mathbb{P}_t^{\pi_1}(\boldsymbol{s}', a'|\boldsymbol{s}^*, 0) R(\boldsymbol{s}', a', \boldsymbol{s}' \circ a')$$

$$< 0.$$

Here, the last equality is due to $\sup_{a>0, a \in \mathcal{A}} |f(\boldsymbol{s}^*, a, \boldsymbol{s}^* \circ a)| \leq \lambda \leq \inf_{a<0, a \in \mathcal{A}} |f(\boldsymbol{s}^*, a, \boldsymbol{s}^* \circ a)|$. Hence, we have $\Delta_a < 0$ for $a \in \mathcal{A} \setminus \{0\}$. The last inequality holds by Assumption 1. Therefore, we have $Q^{\pi_1}(\boldsymbol{s}^*, 0) < Q^{\pi_0}(\boldsymbol{s}^*, 0) = 0$, which means $\pi_1$ is not the optimal one. So, we have $\pi^*(0|\boldsymbol{s}^*) = 1$. This completes the proof for the first part.

For the second part, suppose we have a specific state $\boldsymbol{s} \neq \boldsymbol{s}^*$ where $\pi^*(0|\boldsymbol{s}) = 1$. Then, similar to the proof in the first part, we have $V^{\pi^*}(\boldsymbol{s}) = Q^{\pi^*}(\boldsymbol{s}, 0) = 0$. Then, based on the definition of $\pi^*$, $V^{\pi}(\boldsymbol{s}) \leq V^{\pi^*}(\boldsymbol{s}) = 0$ and $Q^{\pi}(\boldsymbol{s}, 0) \leq Q^{\pi^*}(\boldsymbol{s}, 0) = 0$ hold for any $\pi \in \Pi$. Note that in our problem, the optimal state $\boldsymbol{s}^*$ is unique. Hence, there exists a path $\mathcal{D}$ with finite length so that $\boldsymbol{s}$ can be transit to $\boldsymbol{s}^*$, i.e., $\boldsymbol{s} \rightarrow \cdots \rightarrow \boldsymbol{s}^*$. We use $d_0$ to denote the length of the path. Following the path $\mathcal{D}$, we construct a deterministic policy $\pi'$ so that $\boldsymbol{s}$ can transit to $\boldsymbol{s}^*$ with probability equal to 1, and $\pi'(0|\boldsymbol{s}^*) = 1$. Let $\tilde{\boldsymbol{s}}_t$ be the $t$-step state in this path. Now we evaluate $Q^{\pi'}(\boldsymbol{s}, 0)$. Since $\pi'$ is a deterministic policy, we have

$$Q^{\pi'}(\boldsymbol{s}, 0) = 0 + \sum_{t=1}^{d_0} \gamma^t R(\tilde{\boldsymbol{s}}_t, \pi'(\tilde{\boldsymbol{s}}_t), \tilde{\boldsymbol{s}}_t \circ \pi'(\boldsymbol{s}_t)) + \gamma^{d_0+1} V^{\pi'}(\boldsymbol{s}^*) > 0,$$

which is contradictory with $Q^{\pi'}(\boldsymbol{s}, 0) \leq 0$. This completes the proof.

# C   Appendix C: Additional Experiment Results

## C.1   Additional Experiment Results for Linear Regression

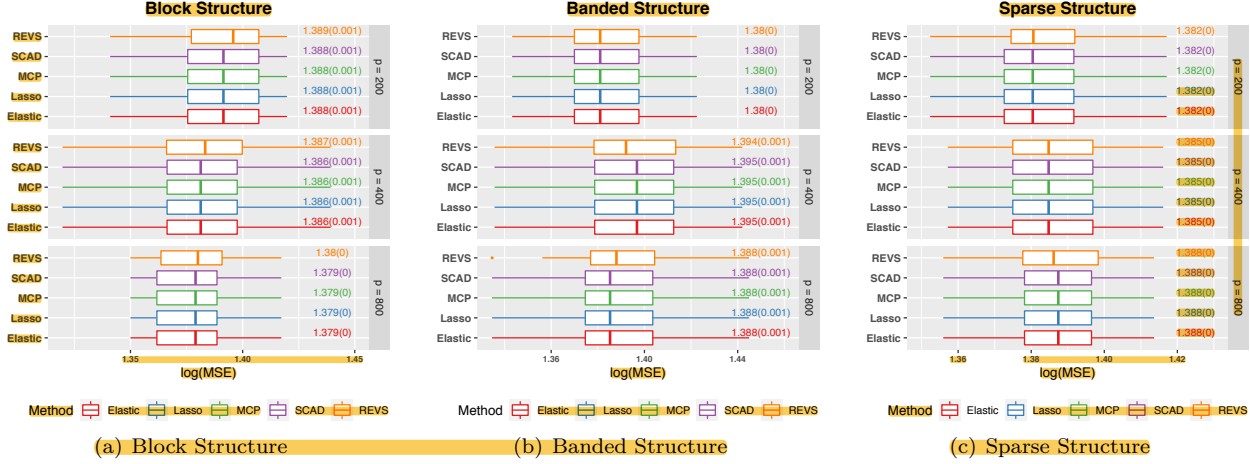

(a) Block Structure          (b) Banded Structure          (c) Sparse Structure

Figure 5: Mean Square Errors (MSE) for predicting $Y$ on testing data in *linear regression* with *big data setting*. The statistics in the text show the mean and variance of MSE across 50 replications.

## C.2 Additional Experiment Results for Logistic Regression

| | $p = 200$ | | | $p = 400$ | | | $p = 800$ | | |
| --- | --- | --- | --- | --- | --- | --- | --- | --- | --- |
| | # | TNR(%) | PPV(%) | # | TNR(%) | PPV(%) | # | TNR(%) | PPV(%) |
| **Block Structure** | | | | | | | | | |
| Lasso | 16.70 | 98.76% | 78.60% | 19.98 | 98.28% | 67.56% | 25.42 | 98.35% | 49.57% |
| Elastic | 45.06 | 83.64% | 39.01% | 48.26 | 91.28% | 35.57% | 72.12 | 92.60% | 25.29% |
| MCP | 16.98 | 98.13% | 87.15% | 18.40 | 98.97% | 79.13% | 21.36 | 98.94% | 61.03% |
| SCAD | 22.68 | 95.78% | 67.54% | 29.36 | 96.22% | 51.98% | 41.02 | 96.57% | 35.15% |
| REVS | **16.26** | **99.04%** | **89.73%** | **16.62** | **99.16%** | **81.16%** | **18.96** | **99.03%** | **61.29%** |
| **Banded Structure** | | | | | | | | | |
| Lasso | 16.93 | 98.69% | 77.42% | 21.32 | 97.56% | 64.78% | 23.32 | 98.86% | 51.67% |
| Elastic | 42.16 | 85.94% | 41.31% | 47.74 | 91.87% | 36.86% | 69.13 | 93.35% | 27.21% |
| MCP | 24.73 | 94.76% | 64.26% | 26.31 | 97.02% | 59.13% | 26.78 | 96.68% | 48.83% |
| SCAD | 31.32 | 91.38% | 57.56% | 34.78 | 93.83% | 50.12% | 44.67 | 93.48% | 33.64% |
| REVS | **16.33** | **99.02%** | **91.73%** | **16.86** | **99.13%** | **80.72%** | **18.47** | **99.12%** | **61.34%** |
| **Sparse Structure** | | | | | | | | | |
| Lasso | 18.56 | 97.13% | 75.48% | 21.89 | 97.04% | 64.15% | 24.92 | 98.54% | 50.48% |
| Elastic | 48.07 | 82.76% | 36.84% | 48.77 | 91.56% | 36.08% | 70.36 | 92.88% | 26.92% |
| MCP | 21.37 | 96.83% | 72.15% | 28.40 | 96.31% | 60.72% | 31.46 | 97.37% | 43.12% |
| SCAD | 30.76 | 91.89% | 59.91% | 39.36 | 91.07% | 47.63% | 47.02 | 92.05% | 31.08% |
| REVS | **16.29** | **99.02%** | **91.75%** | **18.97** | **99.12%** | **80.47%** | **19.96** | **98.79%** | **59.83%** |

Table 3: Means of numbers of selected variables (#) with $p_0 = 15$, Positive Predictive Values (PPV), and True Negative Rates (TNR) in *high-dimensional setting* for *logistic regression* under 50 replications. Best values in bold.

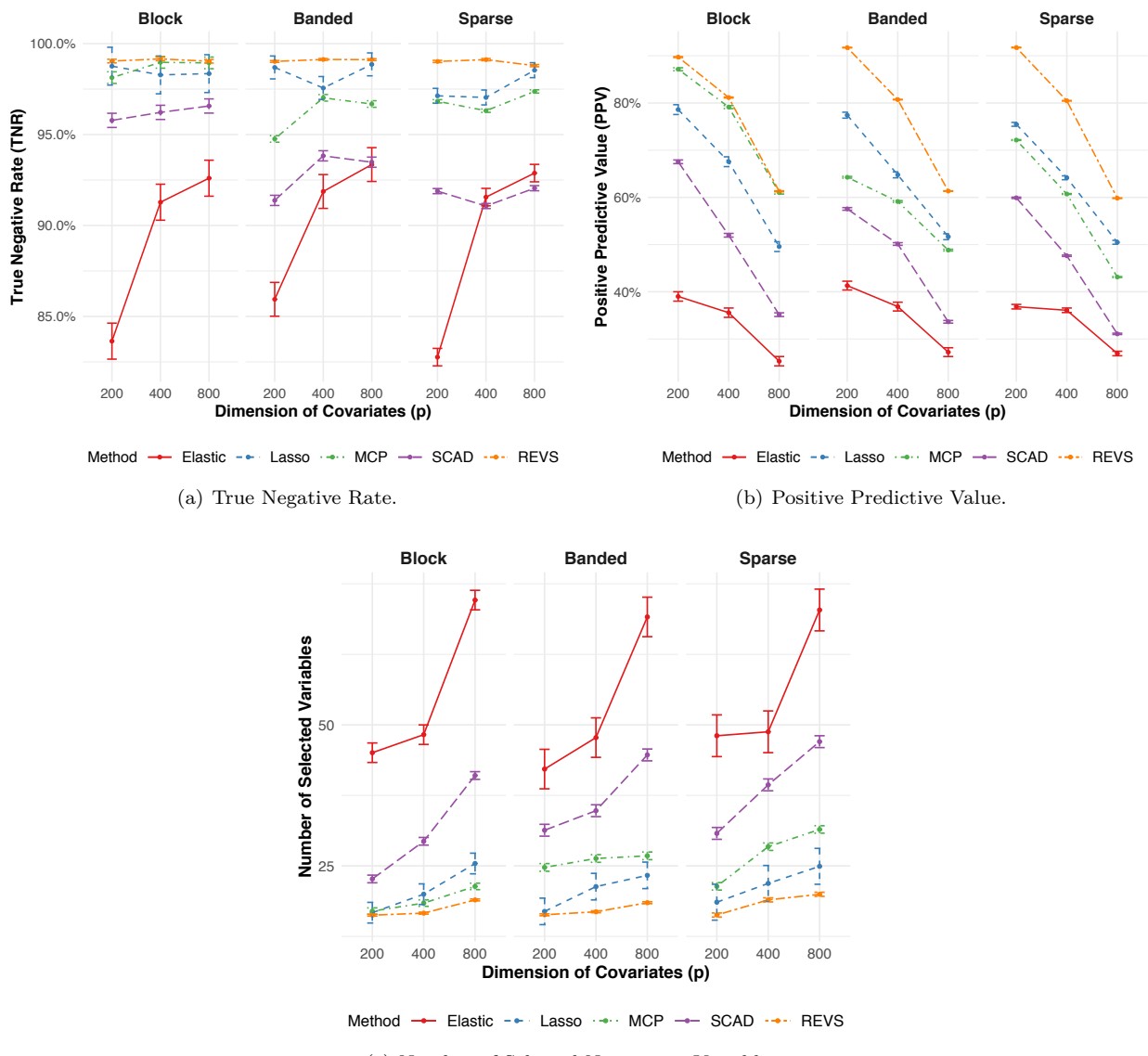

(a) True Negative Rate.

(b) Positive Predictive Value.

(c) Number of Selected Non-sparse Variables.

Figure 6: True Negative Rates (TNR), Positive Predictive Values (PPV), and numbers of selected non-sparse variables of comparison methods in *logistic regression* with *high-dimensional data setting*. The statistics in the text show the mean and variance of above metrics across 50 replications.

## C.3   Additional Experiment Results for Comparison with Divide and Conquer Methods

To compare our method with other regularization methods using small batches, we conduct additional experiments to compare the computation time for linear regression as the example. Since the other methods do not incorporate an MDP design, we used the divide-and-conquer (D&C) approach for comparison. Specifically, we divided the entire dataset into 400 batches, mimicking the training sample size used by REVS in each batch. For each batch, we fit penalized regression models to select variables with non-sparse estimated coefficients, employing 5-fold cross-validation to tune the parameters. Finally, we averaged the variable selection frequency across the 400 batches and applied a threshold of 0.6 to construct the final selected variable set. These experiments were replicated 50 times under the same big data settings described in Section 5 (entire dataset size = 500,000), considering the same three different precision matrix structures

for the features. An independent test set of 5,000 samples was used to evaluate performance. The mean computation times (in minutes) are summarized in the table below:

Table 4: Means of computation time (minutes) in *big data setting* under 50 replications. We compare our REVS method with other Divide and Conquer (D&C) methods. Best values in bold.

| Structure | Method | $p = 200$ | $p = 400$ | $p = 800$ |
|---|---|---|---|---|
| Block $\Omega$ | Lasso (D&C) | 3.18 | 6.51 | 18.12 |
| | Elastic (D&C) | 6.88 | 17.02 | 52.65 |
| | SCAD (D&C) | 1.70 | 4.34 | 62.34 |
| | MCP (D&C) | 1.78 | 4.68 | 69.65 |
| | REVS | **0.34** | **0.82** | **3.38** |
| Banded $\Omega$ | Lasso (D&C) | 6.87 | 10.91 | 11.09 |
| | Elastic (D&C) | 14.95 | 26.05 | 31.25 |
| | SCAD (D&C) | 4.92 | 16.63 | 56.83 |
| | MCP (D&C) | 5.28 | 18.32 | 69.41 |
| | REVS | **0.60** | **1.82** | **2.78** |
| Sparse $\Omega$ | Lasso (D&C) | 5.28 | 6.78 | 19.74 |
| | Elastic (D&C) | 11.44 | 17.34 | 59.35 |
| | SCAD (D&C) | 3.89 | 9.09 | 69.33 |
| | MCP (D&C) | 4.05 | 9.52 | 78.90 |
| | REVS | **0.40** | **0.81** | **4.04** |

