# OpenReview forum: "Efficient Variable Selection Using Reinforcement Learning for Big Data"
_TMLR — Rejected by TMLR_

### Review · Reviewer_hUcd · 2024-12-09

**Summary Of Contributions:**

The paper proposes a novel approach called REinforcement learning for Variable Selection (REVS) to address variable selection challenges in big data. It formulates the problem as a Markov Decision Process (MDP) and uses RL techniques. In experiments, REVS shows superior computational efficiency compared to methods like Lasso, Elastic net, SCAD, and MCP, while the all methods have similar accuracy.

**Audience:**

Yes

**Claims And Evidence:**

Yes

**Requested Changes:**

Please explain more about what is new in this paper conparing with previous work on "feature selection by RL".

**Strengths And Weaknesses:**

Strenths:
1. Clear MDP formulation and proper application of RL, with concrete theoretical proofs.
2. Good presentation.

Weaknesses:
1. In the setting, the Generalized Linear Model is adopted, and in the experiments only linear regression and logistic regression is considered. I think the proposed REVS should be a more general framework, suitable for many other supervised learning tasks.
2. REVS, although named as "variable selection", seems to more related to the field of "feature selection". The MDP formulation and TD algorithm are kind of straighforward. Although there presented several papers on feature selection by RL, it is not clear what is the most important innovation of this work.

---

> ### Author Response · Authors · 2025-01-02
> **Response to Review**
>
> Thank you for taking the time to review our work and for providing insightful and constructive feedback. We greatly appreciate your effort in evaluating our paper. Please see the following response to the review.
>
> **Response to Weakness 1**:
>
>  Thank you for your insightful comments on the use of GLM in our framework. Since our immediate reward function is defined based on the improvement in model fit, our REVS approach is indeed generalizable and not limited to linear and logistic regression. In this initial study, we focused on these models due to their prevalence and well-understood properties. We recognize the importance of extending the experiments to other GLM types, such as multinomial and Poisson regression, to address diverse research questions. We will also explore applications beyond the GLM family to broaden the framework's scope and applicability in future work.
>
> **Response to Weakness 2**:
>
> Thank you for your feedback and observation in terms of the distinction between variable selection and feature selection. While our method is named "variable selection", it is indeed applicable to both variables in the original dataset and derived features. As long as the set of variables or constructed features is determined for modeling, REVS can effectively identify the most relevant subset of them.
>
> In addition, our proposed REVS introduces significant advancements that address big data challenges compared with traditional variable selection techniques. (1) Our approach offers computational efficiency by decomposing the variable selection process into sequential actions, each involving small data subsets, and fitting simple GLM models. This design drastically reduces computational load and memory usage, making REVS highly effective for large datasets. (2) REVS incorporates an $\epsilon$-greedy algorithm to balance exploration and exploitation, avoiding local optima and ensuring robust feature selection over multiple batches. (3) The batch-wise dynamic adjustment also helps mitigate overfitting by leveraging cumulative reward information, ensuring variable selection accuracy and generalizability especially for high dimensional data with highly correlated features.
>
> Compared to RL-based feature selection methods, such as those in [1,2], our work differs in key aspects, including a more flexible MDP design that allows both adding and removing variables, and a generalized reward function based on log likelihood. These innovations, along with the theoretical guarantees in Theorem 1, distinguish REVS as a robust framework for feature selection. For a detailed comparison with existing RL-based methods, please refer to the overall response provided to all the reviewers, where these differences are discussed comprehensively.
>
> **Response to Requested Changes**:
>
> Thank you for your feedback and suggestions for highlighting the importance of distinguishing our work from previous studies on "feature selection by RL". As noted in our above combined response, our method introduces key innovations, including a more flexible MDP design that allows both adding and removing variables, a generalized reward function based on log likelihood, and theoretical guarantees such as Theorem 1.
>
> We have also added detailed comparisons with existing RL-based methods [1,2], directly into our paper. These differences and advancements are discussed in detail in the fourth part of Section 1.3.
>
> *References*:
>
> [1] Rasoul, Sali, Sodiq Adewole, and Alphonse Akakpo. "Feature selection using reinforcement learning." arXiv preprint arXiv:2101.09460 (2021);
>
> [2] Fard, Seyed Mehdi Hazrati, Ali Hamzeh, and Sattar Hashemi. "Using reinforcement learning to find an optimal set of features." Computers & Mathematics with Applications 66.10 (2013): 1892-1904.

---

### Review · Reviewer_zhSV · 2024-12-16

**Summary Of Contributions:**

In this work, the authors propose a reinforcement learning-based algorithm for efficient variable selection in machine learning models (REVS). The framework employs an iterative search process where, at each iteration, the set of variables is evaluated and adjusted using an $\epsilon$-greedy policy update. The authors provide a theoretical justification for their method and present experimental results demonstrating its computational efficiency, particularly in terms of reduced runtime.

**Audience:**

No

**Claims And Evidence:**

No

**Requested Changes:**

- In Subsection 5.1, the authors mention that "all methods exhibited similar results in terms of TNR, PPV, and MSE for variable selection accuracy." However, these results are not explicitly reported. I request that the authors provide these metrics, explicitly measured on a withheld test set that was not accessed during the execution of the methods.
- I ask the authors to clarify how state values $V$ and $Q$-values are stored during the run of the algorithm. Given that the state space is of size $2^p$, storing them in a tabular manner does not seem feasible.
- I ask the authors to evaluate their method on a commonly used real-world dataset with a standard train/validation/test split and report its performance on this dataset.
- To ensure reproducibility, I strongly encourage the authors to provide their code along with the paper.

**Strengths And Weaknesses:**

I apologize in advance if any of the following questions or critiques arise from a misunderstanding of the paper. However, I have several significant concerns, which I have outlined below. I kindly ask the authors to address these points and provide clarifications. I am open to revising my decision if satisfactory responses are provided.

#### Conceptual Concerns:
- I find the core idea of the paper unclear. The problem of optimal variable selection is inherently an optimization problem on a multidimensional grid. Introducing sequential decision-making and framing the problem as an MDP seems unnecessary and does not appear to provide meaningful advantages over standard optimization techniques.

#### Theoretical Concerns:
- Lemma 1 seems to be a restatement of the definitions of $V $ and $Q$ rather than a genuine lemma. It does not provide any new insight or claim and therefore appears superfluous.
- In Subsection 3.2, the discussion of computational efficiency mentions that estimations are computed on a subset of the data. Does the theoretical analysis account for this? If so, Assumption 1 appears unfeasible, and the proof of Theorem 1 should explicitly reflect this. If the analysis assumes the full dataset is used, then Assumption 1 always holds, and Theorem 1 becomes trivial.

#### Experimental Concerns:
- The comparison against Lasso, Elastic Net, SCAD, and MCP is problematic. These regularization methods are primarily designed to prevent overfitting and improve performance on unseen data, not necessarily to identify the correct predictors. Comparing REVS to these methods seems invalid. I recommend adding comparisons against more relevant methods, such as grid search approaches (e.g., random grid search).
- The performance evaluation, based primarily on computation time, is unfair. The artificially created dataset used in the experiments is disproportionately large for the problem at hand. Competing methods are forced to operate on the entire dataset, whereas REVS benefits from processing only a small subset. This design inherently favors REVS. I suggest re-running the experiments to evaluate the performance of the other methods on smaller subsets of the dataset for a more equitable comparison.

---

### Review · Reviewer_xXUD · 2024-12-19

**Summary Of Contributions:**

This paper solves the variable selection problem by reformulation to a reinforcement learning problem. The authors argue that in the setting of large sample size with highly correlated features, the proposed approach is computationally efficient and shows better performance than the penalized methods including LASSO. Several experiments to verify the arguments are provided and theoretical analysis of the algorithm is provided.

**Audience:**

Yes

**Claims And Evidence:**

Yes

**Requested Changes:**

1. In page 3, $X$ is defined as a column vector but in section 2.1, it is now defined as set of features, and $2^X$, where $X$ seems to be a set. I would recommend using a different notation for set and a vector.


2. There is a typo in page 19: Corolary 1.5 -> Corollary 1.5.

3. I would recommend adding the theorem and corollaries in the cited papers for the completeness of the paper, e.g., Corollary 1.5 and Lemma 1.6 in Agarawal et al., and so on.

4. Can the authors give more comment on why the proposed method is effective for highly correlated data?

**Strengths And Weaknesses:**

**Strength**


1. The proposed approach can provide new perspective on the variable selection problem. The detailed MDP formulation is insightful and will be likely to helpful to the community.

2. The paper theoretically establishes conditions on the regularization coefficient on the reward function for the proposed algorithm to converge.

3. The claim of the paper, that the proposed method is computationally efficient than penalty-based methods, is experimentally verified.

**Weakness**

1. The advantage of computational complexity is not clear in theoretical sense. Can the authors provide more comments on why the proposed method has efficient computation time, e.g. in terms of $n_{train}$ or $p$? Moreover, the proposed approach requires additional step of learning procedure of the RL algorithm.

2. I have some concerns about the experiment setting. How are the coefficients for REVS, LASSO and other algorithms selected? The experimental results will be stronger if the authors can provide the result for a wide range of regularization coefficients. Moreover, why did the authors exclude tree-based methods in their comparison?

3. A similar approach has been studied in Rasoul et al. [1,2]. A detailed comparison on e.g., choice of reward function, definition of state space would be helpful to understand the difference with existing works.

[1] Rasoul, Sali, Sodiq Adewole, and Alphonse Akakpo. "Feature selection using reinforcement learning." arXiv preprint arXiv:2101.09460 (2021).

[2] Fard, Seyed Mehdi Hazrati, Ali Hamzeh, and Sattar Hashemi. "Using reinforcement learning to find an optimal set of features." Computers \& Mathematics with Applications 66.10 (2013): 1892-1904.

---

### Decision · Action_Editor_Cfqs · 2025-01-26

**Recommendation:** Reject

**Comment:**

**Presentation and Empirical Evidence:**

The presentation could be improved by including more concrete examples or empirical evidence that demonstrates the advantages of the MDP framework over traditional methods. This would help readers better understand the practical benefits of the proposed approach.

Additionally, the authors should discuss the sensitivity of the MDP framework's performance to the design of the reward function. Providing guidelines for defining the reward function in different contexts would be valuable.

**Justification Assumption 1:**

Assumption 1, which is crucial for the proof of the main theory, is quite strong. It assumes that any deviation from the optimal state is extremely detrimental and that the agent might never recover from it. While this assumption may hold in scenarios with a high noise-to-signal ratio where irrelevant features can dominate model performance, the authors should provide a more comprehensive discussion on when this assumption is satisfied in variable selection applications.

Including motivating examples and experimental validation would help clarify the conditions under which Assumption 1 is valid.

**Exploration Strategies:**

The authors should consider adding more exploration strategies to the experimental comparison. While they mention this as a future extension, incorporating basic strategies such as UCB (Upper Confidence Bound), TS (Thompson Sampling), or common RL algorithms like Q-learning would provide a more robust comparison. These additions seem feasible and would enhance the paper's comprehensiveness.

**Additional Experiments in Section C:**

It would be beneficial to include explanations or discussions on the additional experiments presented in Section C. Currently, the authors only provide tables and figures without any analysis or description. Adding context and interpretation would improve the clarity and usefulness of these results.

**Title and Abstract:**

The title and abstract should explicitly mention that the paper focuses on generalized linear models (GLMs), as this is the only model considered. This clarification will prevent any potential overclaim and ensure that readers have a clear understanding of the paper's scope.

Furthermore, the authors should discuss the versatility of their framework and its applicability across various modeling contexts. This discussion would highlight the broader relevance and potential extensions of their work.

**Audience:**

This paper should be intereting to general ML audience who works on reinforcement learning and high dimensional applications.

**Claims And Evidence:**

This paper investigates an intriguing problem by employing reinforcement learning (RL) to enhance the efficiency of variable selection in generalized linear models (GLMs). All reviewers found the problem formulation interesting and the results promising. However, several valid concerns were raised during the review process regarding the validity of the problem formulation, the theoretical assumptions, and the presentation. I encourage the authors to undertake a major revision to validate the correctness and rigor of their formulation and analysis. Addressing these concerns will significantly strengthen the paper and enhance its potential impact.

**Resubmission Of Major Revision:**

The authors may consider submitting a major revision at a later time.